



# Seasonal dynamics and spatial patterns of soil moisture in a loess catchment

Shaozhen Liu[1,2,3], Ilja van Meerveld[4], Yali Zhao[1], Yunqiang Wang[1,6], James W. Kirchner[2,5]

[1]State Key Laboratory of Loess and Quaternary Geology, Institute of Earth Environment, Chinese Academy of Sciences, Xi'an, China

[2]Department of Environmental Systems Science, ETH Zürich, Zürich, Switzerland

[3]Interdisciplinary Research Center of Earth Science Frontier, Beijing Normal University, Beijing, China

[4]Department of Geography, University of Zurich, Zurich, Switzerland

[5]Swiss Federal Research Institute WSL, Birmensdorf, Switzerland

[6]Department of Earth and Environmental Science, Xi'an Jiaotong University, Xi'an, China

*Correspondence to*: Yunqiang Wang (wangyq@ieecas.cn)

**Abstract.** The spatial patterns and temporal dynamics of soil moisture in loess landscapes are not well understood. In this study, volumetric soil moisture was monitored monthly for 6.5 years at 20 cm intervals between the surface and 500 cm depth at 89 sites across a small (0.43 km[2]) catchment on the
Chinese Loess Plateau. The median soil moisture was computed for each month for each monitoring site as a measure of typical soil moisture conditions. Seasonal changes in soil moisture were mainly concentrated in the shallow (0-100 cm) soil, with a clear seasonal separation between wet conditions in October-March and dry conditions in May-July, even though precipitation is highest in July-August. Soil moisture was higher on northwest-facing slopes, due to increased drying from solar radiation on
southeast-facing slopes. This effect of slope aspect was greater between October and March, when the zenith angle of the sun was lower and the aspect-dependent contrast in solar radiation reaching the surface was larger. The wetter, northwest-facing slopes were also characterized by larger annual soil moisture storage changes. Soil texture was nearly uniform across both slopes, and soil moisture was not correlated with the topographic wetness index, suggesting that variations in evapotranspiration



dominated the spatial pattern of soil moisture in shallow soils during both wet and dry conditions.
Water balance calculations indicate that over 90% of annual precipitation was seasonally cycled in the
soil between 0 and 300 cm, suggesting that only a minor fraction infiltrates to groundwater and
becomes streamflow. Our findings may be broadly applicable to loess regions with monsoonal climates,
and may have practical implications for catchment-scale hydrologic modeling and the design of soil
moisture monitoring networks.

**Key words:** soil moisture, spatial patterns, seasonal dynamics, soil moisture storage change, loess
catchment

**1. Introduction**

Understanding the spatial variability of soil moisture is critical to the study of transpiration, streamflow
generation, groundwater recharge, land-atmosphere interactions, and soil ecology and biogeochemistry
(Dymond et al., 2021; Ridolfi et al., 2003), as well as hydrological applications ranging from
streamflow forecasting to irrigation management (Brocca et al., 2010; Chen et al., 2011; Koster et al.,
2010; Peterson et al., 2019). The spatial heterogeneity of soil moisture usually varies with the average
field-, hillslope-, and catchment-scale wetness (Western et al., 2003). Usually the spatial variability of
soil moisture is highest at intermediate average wetness, and lowest at extreme dry or wet conditions
(Choi and Jacobs, 2007; Famiglietti et al., 2008; Kaiser and McGlynn, 2018; Owe et al., 1982;
Rosenbaum et al., 2012; Teuling and Troch, 2005; Western et al., 2003). Spatial patterns of soil
moisture are also shaped by topography, soil properties, and vegetation (Han et al., 2021; Tromp-van
Meerveld and McDonnell, 2006). The influence of these factors varies with soil wetness or seasonality,
due to shifts in the dominant hydrological processes regulating soil moisture (Jarecke et al., 2021;
Liang et al., 2017; Western et al., 2004). Grayson et al. (1997) and Western et al. (2003) demonstrated
that topography has a greater influence on spatial patterns of soil moisture under wet conditions, due
to redistribution of soil moisture by lateral flow, resulting in wetter soils along hillslope drainage lines
in convergent topography. Under dry conditions, by contrast, soil properties and vegetation become
more important factors because soil moisture is mainly affected by point-scale vertical water fluxes.
Any topographic influence under dry conditions is more likely to be due to aspect rather than
topographic convergence (Grayson and Western, 2001). Grayson and Western (2001) summarized this



phenomenon as local and nonlocal control on soil moisture under dry and wet conditions, respectively.

Many studies have attempted to understand spatial patterns in soil moisture and their local and nonlocal controls (Dymond et al., 2021; Hoylman et al., 2019; Jarecke et al., 2021; Kaiser and McGlynn, 2018; McNamara et al., 2005; Penna et al., 2009; Tromp-van Meerveld and McDonnell, 2006; Williams et
al., 2009), sometimes reaching different conclusions than Grayson et al. (1997) and Western et al. (2003). For example, in the Mediterranean climate of Caspar Creek Experimental Watershed, California, USA (annual precipitation 1168 mm, volumetric soil moisture ~20-~40%), Dymond et al. (2021) found that average soil moisture in the wet season did not follow typical topographic drivers, i.e., topographic wetness index (TWI) and upslope accumulated area (UAA). Similarly, at H. J.
Andrews Experimental Forest, Oregon, USA (annual precipitation 2450 mm, volumetric soil moisture ~16-~32%), Jarecke et al. (2021) found that hillslope soil moisture was largely independent of hillslope topography and instead primarily controlled by soil properties, under both wet and dry conditions. At the Hemuqiao Hydrological Experimental Station, southeastern China (annual precipitation 1580 mm, volumetric soil moisture ~20-~40%), Han et al. (2021) found that the relation between volumetric soil
moisture and topography fluctuated as a function of catchment and precipitation characteristics. Relatively few studies have been conducted in arid/semiarid areas. In semi-arid montane catchments at Lubrecht and Tenderfoot Creek Experimental Forests, Montana, USA (Hoylman et al., 2019; Kaiser and McGlynn, 2018), the spatial organization of soil moisture across catchments was persistent over time and strongly influenced by topographic convergence and divergence, even at the end of the
growing season when the catchment was at its driest state. These contrasting observations have been ascribed to site-to-site differences in catchment topography, climate, soil characteristics, and perennial source areas, and thus to differences in the dominant hydrological processes under both dry and wet conditions (Kaiser and McGlynn, 2018; Takagi and Lin, 2011; Western et al., 2004).

Loess catchments, with their relatively uniform subsurface, are ideal locations to study the effects of topography and slope aspect on soil moisture. The Loess Plateau, situated in the middle and upper reaches of China's Yellow River basin, has the largest and deepest loess deposits in the world (Jia et al., 2015; Zhu et al., 2019). Most of the area is characterized by a semi-arid to semi-humid climate, with an average annual precipitation of less than 600 mm, of which most falls during the summer





monsoon season (Wang et al., 2011). Due to the uneven distribution of rainfall between seasons, the
high erodibility of loess soils, and sparse vegetation cover, the region is subject to severe soil erosion,
resulting in a dissected landscape (Huang and Shao, 2019; Wang et al., 2019) that may result in distinct
soil moisture patterns. Several studies have examined soil moisture patterns in catchments on the Loess
Plateau (Gao et al., 2016; Qiu et al., 2001; Wang et al., 2019; Yu et al., 2018; Zhang and Shangguan,

2016), but few have documented how these spatial patterns change seasonally, or how they reflect local
and nonlocal controls.

Understanding the effects of local and nonlocal controls on soil moisture patterns can shed light on
dominant hydrological mechanisms controlling near-surface soil moisture (Kaiser and McGlynn,

2018). Our study aims to examine soil moisture spatial patterns and their controls in a Loess Plateau
catchment, focusing on the following questions:

    1.  At which soil depths do seasonal changes in volumetric soil moisture mainly occur?

    2.  Are there spatial patterns in soil moisture, and do these patterns change seasonally?

    3.  How do local and nonlocal attributes affect soil moisture patterns?

105        4.  How does the variability in soil moisture change as a function of average wetness?

## 2. Study area

Our study was conducted in the 0.43 km$^2$ Gutun catchment, located near the center of the Loess Plateau
(Fig. 1). The climate of the study region is continental monsoon, with hot, wet summers and cool, dry

winters. The 60-year average annual precipitation (1956-2015) is 541 mm/year (and was 556 mm/year
for the 2016-2021 study period), more than half of which falls in summer (accounting for 56% of
annual rainfall in 2016-2021), often accompanied by intense thunderstorms. The average annual
temperature (1956-2015) is 9.8°C. The elevation in the study areas varies from 974 m to 1188 m, and
the slope gradient ranges from 0 to 52°. Since the beginning of the "Gully land consolidation" project

in 2011, the gully in Gutun catchment has been filled and leveled using soil from the slopes, resulting
in slopes near 0° along the gully axis. Apart from the gully, the catchment includes two slopes,
predominantly facing southeast and northwest respectively. Soils in the catchment are predominantly
composed of silty loam, ranging in depth from approximately 3 m (in the gully) to more than 30 m (on
the slopes) and are underlain by thick loess deposits. Vegetation on the slopes is dominated by black





locust (*Robinia pseudoacacia L.*), sea buckthorn (*Hippophae rhamnoides L.*), and silver grass (*Stipa bungeana Trin.*); the gully is mainly used for rain-fed cropland.

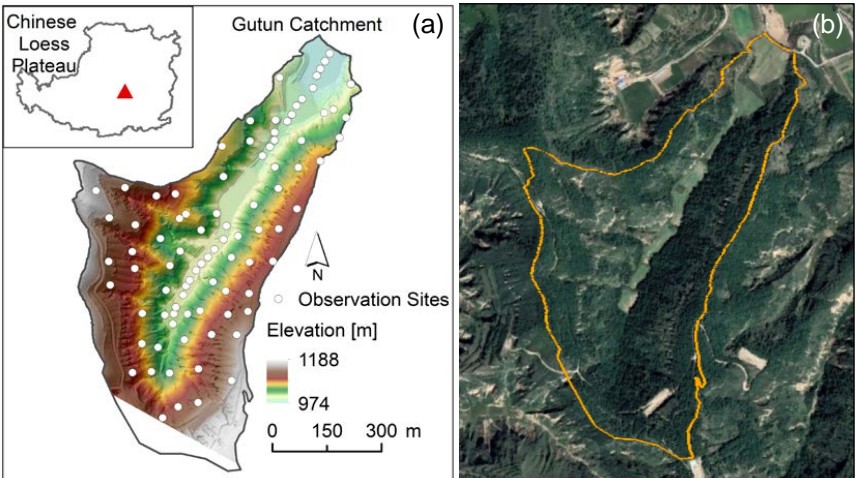

**Fig. 1. Map of the Gutun catchment showing the distribution of the 89 monitoring sites for**
**volumetric soil moisture (a); a satellite imagery from © Google Earth (taken in 2020) showing**
**the relatively lush vegetation on the northwest-facing slope (b). The inset in (a) shows the**
**location of the catchment in the Loess Plateau.**

### 3. Materials and methods

**3.1 Data collection**

Soil moisture was monitored at a total of 89 locations (64 on the slopes and 25 in the gully; Fig. 1). At each monitoring site, soil samples were collected at 20 cm intervals down to a depth of 500 cm using a 5 cm diameter soil auger, except for some gully sites where saturation limited the depth of augering. Each soil sample was air-dried, crushed, and sieved through a 1 mm mesh. The resulting processed
soil samples were then analyzed using the laser diffraction technique (Mastersizer3000, Malvern Instruments, England) to determine the sand, silt, and clay content.

A 500 cm long aluminum neutron probe access tube (CNC100, Probe Technology (Beijing) Co., Ltd, China) was installed vertically into the soil at each of the 89 auger sites. Volumetric soil moisture ($\theta$,
hereafter referred to as soil moisture) was measured monthly at 20 cm intervals from slow-neutron count rates using the revised calibration curve (Wang et al., 2015) based on measurements of the



gravimetric soil moisture and bulk density:

$$\theta = 62.233 \cdot C + 0.9459 \qquad (R^2 = 0.92, p < 0.001) \,, \qquad (1)$$

where $C$ is the slow-neutron count rate. Measurement campaigns were carried out monthly between
April 2016 and October 2021, except during instrument repairs or severe weather that made
measurements impossible, resulting in a total of 57 measurements per location and depth.

A meteorological station has been taking hourly measurements of precipitation, temperature, relative
humidity, solar radiation, and wind speed 2 m above ground at the Gutun catchment since 2016. The
meteorological data from April 2016 to October 2021 (the same period as the soil moisture
measurements) were aggregated daily. Daily potential evapotranspiration (PET) was determined using
the FAO Penman-Monteith equation (https://www.fao.org/land-water/databases-and-software/eto-
calculator/en/), based on these daily meteorological measurements.

**3.2 Data analysis**

In our study, we denote each soil moisture measurement as $\theta_{ijkn}$, meaning the soil moisture content
$\theta$ at monitoring site $i$, month $j$, soil depth $k$, and year $n$. To represent typical soil moisture conditions
and eliminate outliers, we computed the medians for each site $i$, month $j$, and soil depth $k$, over all
sampling years, representing these as $\theta_{ijk}$.

Because of the much higher soil moisture in the gully than on the slopes, in each month $j$ and soil
depth $k$, we also determined average soil moisture for all gully and slope sites, and for the NW- and
SE-facing slopes sites separately:

$$\theta_{jk} = \frac{1}{N} \sum_{i=1}^{N} \theta_{ijk} \quad , \qquad (2)$$

where $N$ is the number of gully ($N$=25), slope ($N$=64), NW-facing slope ($N$=30), or SE-facing slope
($N$=34) sites. We also determined the average soil moisture over depth 0-100 cm for the gully, NW-
facing slope, and SE-facing slope in each month $j$:

$$\theta_{j0-100} = \frac{1}{P} \sum_{i=1}^{P} \theta_{jk} \quad , \qquad (3)$$





where $P=5$ is the number of soil layers over depth 0-100 cm.


### 3.2.1. Seasonal variability in soil moisture

To determine the seasonal changes in soil moisture for each site and depth, we calculated the deviation in the soil moisture for a given month from the annual average (average over 12 months) for that site and depth. Thus, the seasonal deviation in soil moisture for site $i$, soil depth $k$, and month $j$, $\delta\theta_{ijk}$,

was computed as

$$\delta\theta_{ijk} = \theta_{ijk} - \frac{1}{12}\sum_{j=1}^{12}\theta_{ijk} \quad , \tag{4}$$

We similarly determined the average seasonal deviation in soil moisture in the top 100 cm of soil for each site $i$ in month $j$, as:

$$\delta\theta_{ij0-100} = \frac{1}{P}\sum_{k=1}^{P}\delta\theta_{ijk} \quad , \tag{5}$$

where $P=5$ is the number of soil layers over depth 0-100 cm. Then we determined the average seasonal deviation in soil moisture over 0-100 cm soils for gully, NW-facing slope, and SE-facing slope separately in each month $j$:

$$\delta\theta_{j0-100} = \frac{1}{N}\sum_{i=1}^{N}\delta\theta_{ij0-100} \quad , \tag{6}$$

where N is the number of gully (N=25), NW-facing slope (N=30), and SE-facing slope (N=34) sites.


To determine the soil depths at which the seasonal changes in soil moisture were largest, we computed the 20% trimmed standard deviation (TSD, calculated using the R function sd_trim) of the soil moisture at each site and depth ($\sigma_{ik}$) and identified the depth at which it was greatest. We also determined the depth at which seasonal soil moisture changes collapse (i.e., at which $\sigma_{jk}$ converges to

a small value). We defined the collapse threshold as the minimum $\sigma_{ik}$ plus 10% of the difference between the maximum and minimum $\sigma_{ik}$ at each site. The first depth at which $\sigma_{ik}$ was less than this threshold was defined as the depth at which the seasonal changes collapse.

### 3.2.2. Spatial variability in soil moisture at hillslope scale

We quantified the spatial variability in soil moisture on the slopes two different ways. We calculated



how soil moisture at each site deviated from the slope average for the same month and depth; thus the spatial deviation in soil moisture for site $i$, soil depth $k$, and month $j$, $\delta'\theta_{ijk}$, was computed as

$$\delta'\theta_{ijk} = \theta_{ijk} - \theta_{jk} \quad , \tag{7}$$

where $\theta_{jk}$ is the average soil moisture for the slope sites in month $j$ and soil depth $k$, as described

above. The average of this spatial deviation in soil moisture over depths 0-100 cm was calculated for each slope site $i$ in month $j$ as

$$\delta'\theta_{ij0-100} = \frac{1}{P}\sum_{k=1}^{P}\delta'\theta_{ijk} \quad , \tag{8}$$

where $P$=5 is the number of soil layers over depth 0-100 cm.

The overall spatial variability in soil moisture across the slopes, for each month and depth, was quantified using the trimmed standard deviation. The spatial variability of the soil moisture in month $j$ and soil layer $k$, across the slope, $\sigma_{jk}$, was described by the 20% TSD of $\theta_{ijk}$.

### 3.2.3. Annual soil moisture storage change

The annual soil moisture storage change (ΔS) reflects the balance between incoming precipitation (P), evapotranspiration (ET), deeper percolation, and lateral flow. The annual change in soil moisture storage at site $i$ and depth $k$, $\Delta S_{ik}$, was computed as

$$\Delta S_{ik} = \Delta\theta_{ik} \cdot d \cdot 10 \quad , \tag{9}$$

where $\Delta\theta_{ik}$ is the difference in the median soil moisture at site $i$ and depth $k$ between the wettest

and driest month, $d$=20 is the soil thickness for each layer $k$, and the factor of 10 converts this sampling interval to mm. We defined the wettest and driest months (October and June, respectively) as those with the highest frequency of maximum and minimum soil moisture (averaged from 0-500 cm) across all of the sites. Thus, the same "wettest" and "driest" months were used for all monitoring sites, despite some site-to-site differences in seasonal patterns of soil moisture. Lastly, the total soil

moisture storage changes from depth $k_1$ to $k_2$ at sampling site $i$, $\Delta S_i$, can be defined as

$$\Delta S_i = \sum_{k=k_1}^{k_2}\Delta S_{ik} \quad . \tag{10}$$





*3.2.4. Relation to topography*

We selected aspect and TWI as possible topographic controls on the monthly soil moisture patterns
across the catchment. We calculated aspect and TWI from a Digital Elevation Model (DEM) of the
Gutun catchment, produced from an unmanned aerial vehicle LiDAR scan with 0.5 m resolution. The
DEM was smoothed to 10 m resolution to eliminate the effects of microtopography, and TWI and
aspect were determined in the SAGA GIS platform.

We used Spearman rank correlation ($r_s$) to determine the correlation between TWI and the soil moisture
at a location, depth, and month ($\theta_{ijk}$). To determine the effect of slope aspect on soil moisture, we
calculated the incoming solar radiation for each monitoring site using Points Solar Radiation tool in
ArcGIS. We calculated the statistical significance of the difference in both the incoming solar radiation
and slope average soil moisture ($\theta_{jk}$) using one-way ANOVA.


**4 Results and Discussion**

**4.1 Seasonal changes in soil moisture**

Fig. 2 shows the seasonal changes in the average soil moisture ($\theta_{jk}$) for each depth for the slope and
the gully, respectively. In general, the soils in the gully were much wetter than those on the slopes (see
also Fig. 3) due to gravity-driven lateral convergence of near-surface flow (Fan et al., 2019). On the
slope and in the gully, soil moisture varied seasonally in the shallow soils but remaining roughly
constant in the deeper soils (Figs. 2-3).

On the slopes, shallow soils were wetter than deep soils, on average, from November to January, and
drier than deep soils from May to July (Fig. 2a), although it is worth noting that for 53 out of the 64
slope sites, the shallow soils remained wetter than the deeper soils, even though they tended to dry
down from May to July. In the gully by contrast, the deep soils remained wetter than the shallow soils
throughout the seasons (Fig. 2b). This is also seen in the vertical patterns of the average moisture
content ($\theta_{jk}$) in the slopes and gully (Fig. 3). Soil moisture on the slopes varied more with the depth
in the wettest month than in the driest month (Fig. 3a). In the wettest month, average soil moisture on
the slopes was ~12% at the surface (20 cm), then increased to ~15% at a depth of 60 cm, followed by
a gradual decrease to ~11% at a depth of 240 cm and a slight increase to ~13% from 240 cm to 500



cm depth. By contrast, in the driest month, the average soil moisture on the slopes steadily increased

with depth, from ~7% at the soil surface to ~12% at 500 cm. The vertical pattern of soil moisture in

the gully was similar in the wettest and driest months, showing a sharp increase (20-40 cm), slight

decrease during some months (40-140 cm), and slight increase (140-500 cm) with depth. The moisture

content at the surface was ~20%, increasing to ~40% in deep soils (Fig. 3b).

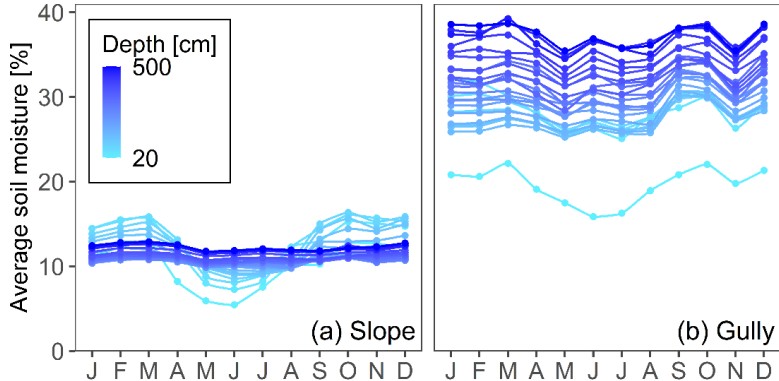

**Fig. 2. Seasonal changes of the average volumetric soil moisture ($\theta_{jk}$) for slope and gully sites. The light blue and dark blue colors indicate the average soil moisture in shallow and deep soils, respectively. The soils in the gully were much wetter than those on the slopes. The rank order of soil moisture with depth reversed between winter and summer on the slopes, but exhibited little seasonal variation in the gully.**




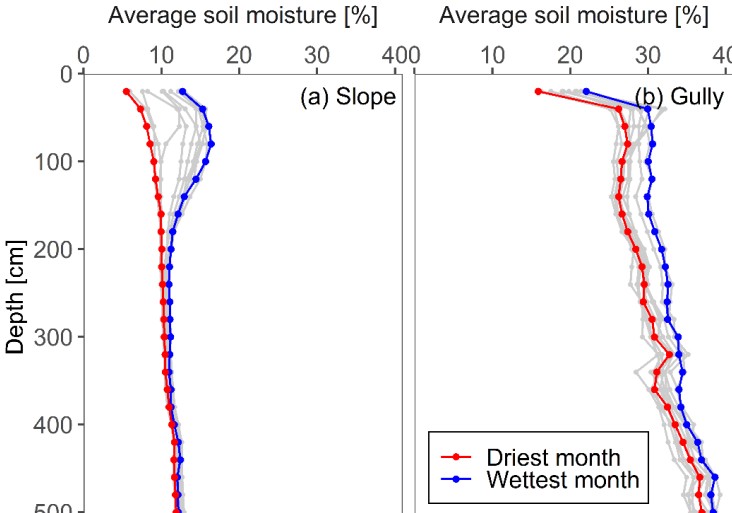

**Fig. 3. The vertical patterns of average volumetric soil moisture ($\theta_{jk}$) on the slope (a) and in the gully (b). The blue and red lines show the soil moisture profiles for the wettest and driest months, respectively. The light grey lines show the profiles for the other ten months. The soil moisture on the slopes varied more as a function of depth in the wettest month than in the driest month, while the profiles in the gully were almost equally steep in the wettest and driest months. Seasonal variations in average monthly soil moisture content were almost exclusively confined to the upper 260 cm on the slopes, but persisted over the full range of depths in the gully.**

For 90% of the sites across the catchment, the depth of the maximum seasonal change in soil moisture (i.e., the maximum $\sigma_{ik}$) was located between 20 and 100 cm (Fig. 4a). The depth at which it collapses (as defined in Sect. 3.2.1) was located between 160 cm and 260 cm for 82% of the sites (Fig. 4b). This suggests that the seasonal variation in soil moisture in the Gutun catchment is largest for 0-100 cm soils, and that there is little seasonal variation below 260 cm. These results are consistent with the findings of several previous studies in the Loess Plateau (Fu et al., 2018; Wang et al., 2019; Wang et al., 2010; Zhao et al., 2020). In addition, we tallied histograms of the months in which the annual maximum and minimum soil moisture occurred for each soil layer within the top 100 cm (Fig. 5). For the 20-60 cm soils, the maximum soil moisture occurred mainly between October to March (but not in January), while at 80-100 cm depth it occurred mainly between September and December (Fig. 5a). The minimum soil moisture values occurred mainly in May, June, and July, regardless of soil depth (Fig. 5b).

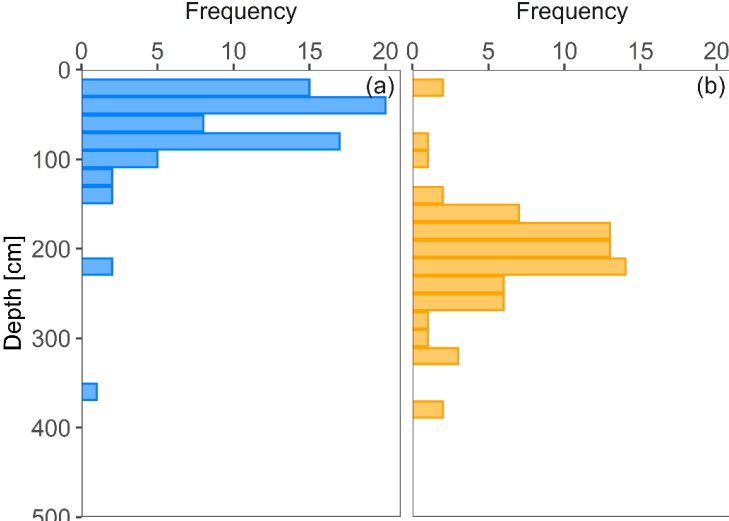

**Fig. 4. Depth of maximum trimmed standard deviation ($\sigma_{ik}$) (a) and depth at which the trimmed standard deviation ($\sigma_{ik}$) collapses, i.e., converges to a small value (b). The results are shown for 72 monitoring sites. The remaining 17 monitoring sites were excluded because they had several null values for deeper soils, making it impossible to calculate the depth at which the trimmed standard deviation collapses. The depth of maximum trimmed standard deviation was between 20 and 100 cm for 90% of the sites. The depth at which it collapses was between 160 cm and 260 cm for 82% of the sites.**

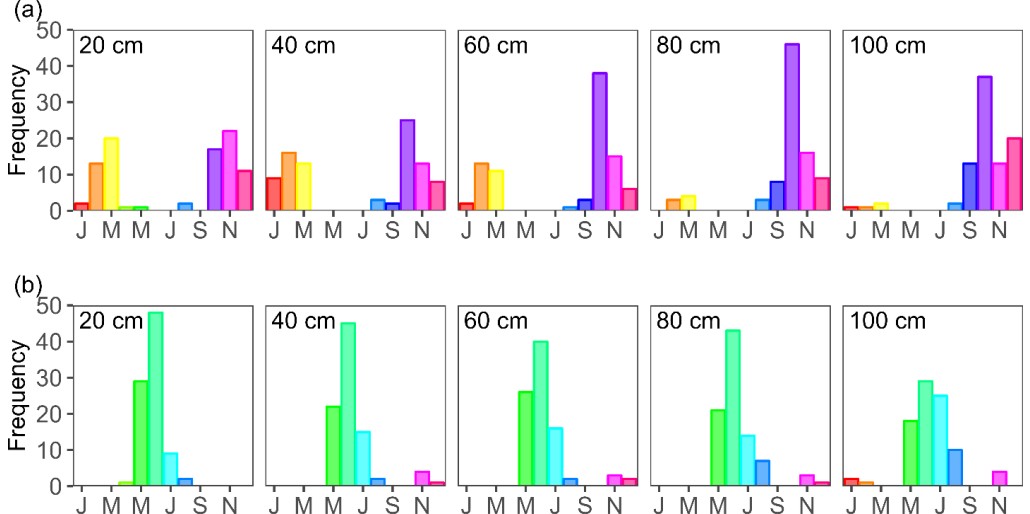

**Fig. 5. Histograms of the month in which the maximum (a) and minimum volumetric soil moisture ($\theta_{ijk}$) (b) occurred in each soil layer within the top 100 cm, based on all 89 monitoring sites across the catchment. At 20-60 cm depth, the maximum soil moisture occurred mainly between October and March (but not in January), while at 80-100 cm depth it occurred mainly between September and December. The minimum soil moisture values occurred mainly in May, June, and July, regardless of depth.**

Figure 6 illustrates the seasonal patterns in average soil moisture content over the top 100 cm for the slope and gully sites ($\theta_{j0-100}$, Fig. 6a), and the deviations from their annual averages ($\delta\theta_{j0-100}$, Fig. 6b). Average soil moisture was much higher in the gully than on the slopes, and the northwest-facing (NW-facing) slope was wetter than the southeast-facing (SE-facing) slope throughout the year (Fig. 6a). The seasonal cycle was also larger on the NW-facing slope than on the SE-facing slope, and was smallest in the gully (Fig. 6b). Thus, seasonal changes in soil moisture were more pronounced on the slopes, especially the wetter (i.e., NW-facing) slope.

Similar to the findings of Grayson et al. (1997), two dominant conditions for 0-100 cm soil moisture were identified: wet (October to March) and dry (May to July), with wet-to-dry transitions around April and dry-to-wet transitions between August and September (Fig. 6a). Together with Fig. 5, these results suggest that soil moisture in the top 100 cm of soil was at a minimum in the late spring and early summer, increasing to a maximum during mid-autumn and remaining relatively wet until early



spring. The period during which the 0-100 cm soils wetted up most rapidly (from July to October; Fig. 6a) coincided with the months in which P exceeded PET (Fig. 6b). The period of soil dry-down (from

March to June; Fig. 6a) also coincided with the months during which PET exceeded P by the largest margin (Fig. 6b). These results are consistent with many studies worldwide that have found an association between seasonal patterns in soil moisture and imbalances between P and PET (Dymond et al., 2021; McNamara et al., 2005; Peterson et al., 2019; Singh et al., 2019; Tromp-van Meerveld and McDonnell, 2006; Williams et al., 2009), even though the soil moisture content and the duration of the

wet and dry states at our site differed markedly from those in previous studies.

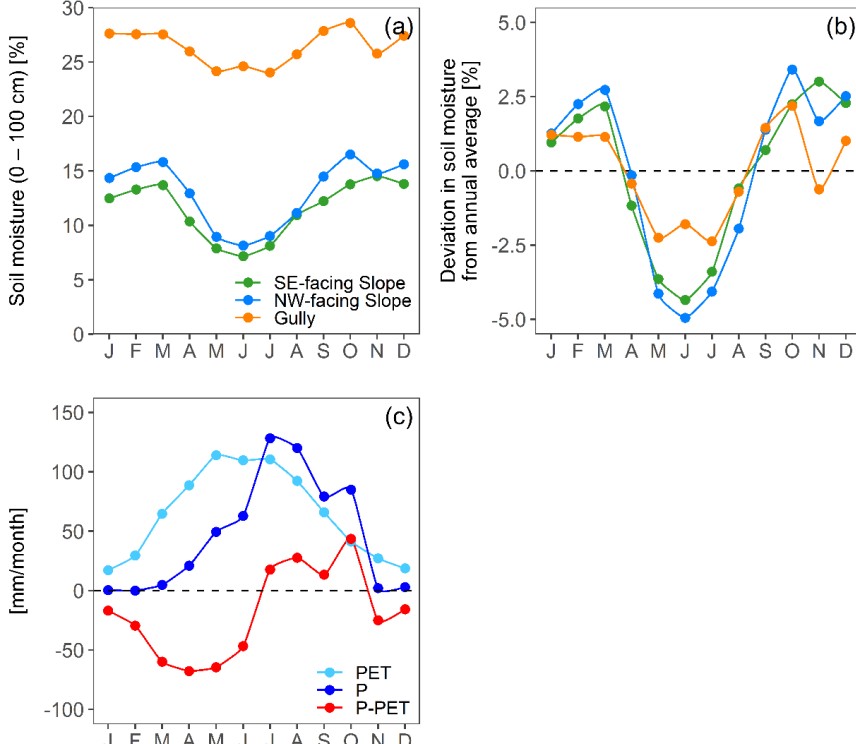

**Fig. 6.** Seasonal changes in average volumetric soil moisture in the top 100 cm of soil for slope and gully sites ($\theta_{j0-100}$, **a**) and the deviations from their annual averages ($\delta\theta_{j0-100}$, **b**). Seasonal patterns in potential evapotranspiration (PET), precipitation (P), and precipitation minus potential evapotranspiration (P-PET) in Gutun catchment (**c**). Soil moisture is much higher in the gully than on the slopes, with the NW-facing slope being wetter than the SE-facing slope throughout the year (**a**). The amplitude of the seasonal change in average soil moisture is largest on the NW-facing slope, followed by the SE-facing slope and the gully (**b**). Soils wetted up most rapidly from July to October (**b**), coinciding with months in which P exceeded PET (**c**). Soil dry-down occurred between March and June (**b**), coinciding with months in which PET exceeded P by the largest margin (**c**).

**4.2 Spatial pattern in soil moisture**

In general, the spatial variation in soil moisture in the top 100 cm of soil on the slopes was smaller (lighter colors) during dry conditions from May to July, and larger (darker colors) during wet conditions from October to March (Fig. 7). This suggests that soil moisture was more homogenous under dry conditions and more heterogeneous under wet conditions (as also shown in Fig. 3). This is consistent with the roughly linear relationships between the spatial TSD ($\sigma_{jk}$, as a measure of spatial





variability) and the spatial average soil moisture ($\theta_{jk}$) for the slope sites (Fig. 8). At each depth within

the top 100 cm of soil, and also for the profile average, the TSD increased linearly with increasing

average soil moisture ($R^2$>0.57).

The higher spatial heterogeneity during wetter conditions may seem inconsistent with the widely

accepted convex-upward model, in which the spatial variability peaks at intermediate values of

moisture content (Brocca et al., 2010; Famiglietti et al., 2008; Jarecke et al., 2021; Tague et al., 2010;

Western et al., 2003). However, that peak often occurs at average moisture values of approximately

20% (Peterson et al., 2019; Shi et al., 2014). We observe a roughly linear, rather than peaked,

relationship between average soil moisture and its variability because the moisture values were mainly

concentrated between 5 and 15% (Fig.2a and Fig. 8). In other words, we may only be observing the

short rising segment of the convex parabola, below the variability peak observed elsewhere.



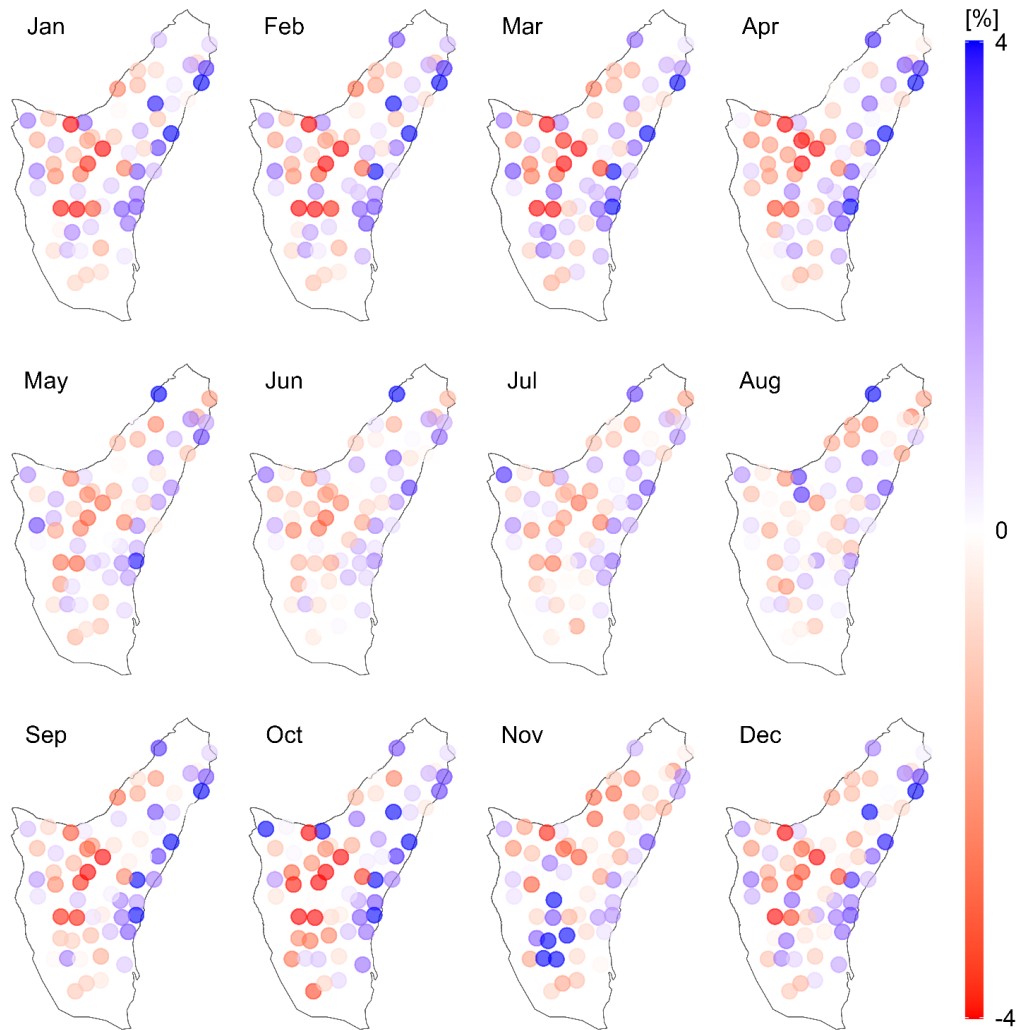

**Fig. 7. Volumetric soil moisture in the top 100 cm of soils at the slope sites: deviation from monthly average across slope sites ($\delta'\theta_{ij0-100}$). The spatial variation in moisture content was smaller (lighter colors) during the dry conditions from May to July, and larger (darker colors) during the wet conditions from October to March. The NW-facing slope was wetter (blue colors), on average, than the SE-facing slope during both wet and dry conditions.**



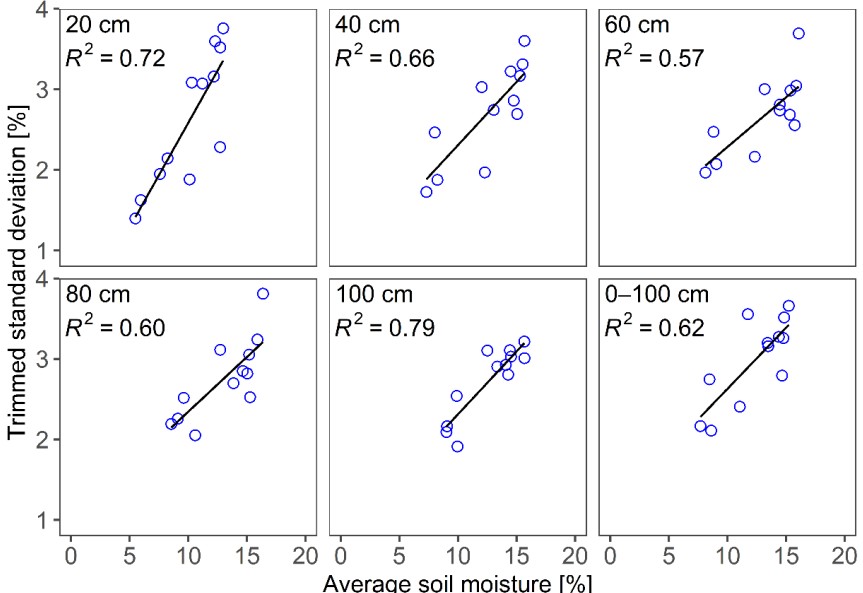

**Fig. 8. Relationships between monthly spatial trimmed standard deviation ($\sigma_{jk}$) and monthly
average volumetric soil moisture ($\theta_{jk}$) at slope sites, for each depth within the top 100 cm of soil
and for the profile average; each point represents a different month of the year. The lines were
fitted using a simple linear regression ($R^2$: 0.57-0.79). The trimmed standard deviation increased
roughly linearly with increasing average soil moisture across slope sites.**


Figure 7 also reveals a spatial pattern in soil moisture, with the NW-facing slope being much wetter

than the SE-facing slope, which was also seen in Fig. 6a and is quantified in more detail in Fig. 9.

During wet months (October to March, except for November), the NW-facing slope is markedly wetter

than the SE-facing slope, while during dry months (May to July) the difference is less distinct (Fig.

9a). The observed pattern is consistent with seasonal differences in solar radiation reaching the two

slopes (Fig. 9b). The topography of the catchment creates variations in local solar angle, and thus in

the total solar radiation received at the surface, leading to topographically driven variations in soil

drying (Fan et al., 2019; Hoylman et al., 2019; Pelletier et al., 2018; Williams et al., 2009). During the

summer months, the higher solar angle in the northern hemisphere weakens the effect of aspect on

solar radiation reaching the surface, leading to smaller differences in evaporation and thus more

consistent soil moisture between the two slopes at the Gutun catchment.

In contrast to findings elsewhere (Geroy et al., 2011), the difference in soil moisture content between



the NW-facing and SE-facing slopes is unlikely to be driven by differences in soil texture and related

differences in water retention. Soil texture at the loess catchment is highly uniform. Across 64 sites on

both slopes, the coefficients of variation for 0-100 cm average clay and silt content were only 0.15 and

0.07, respectively, and the average clay and silt contents in the 0-100 cm soils of the NW-facing slope

were less than 1% higher than those of the SE-facing slope. Thus we do not think that spatial variations

in soil properties are an important driver of the soil moisture spatial patterns at the Gutun catchment.

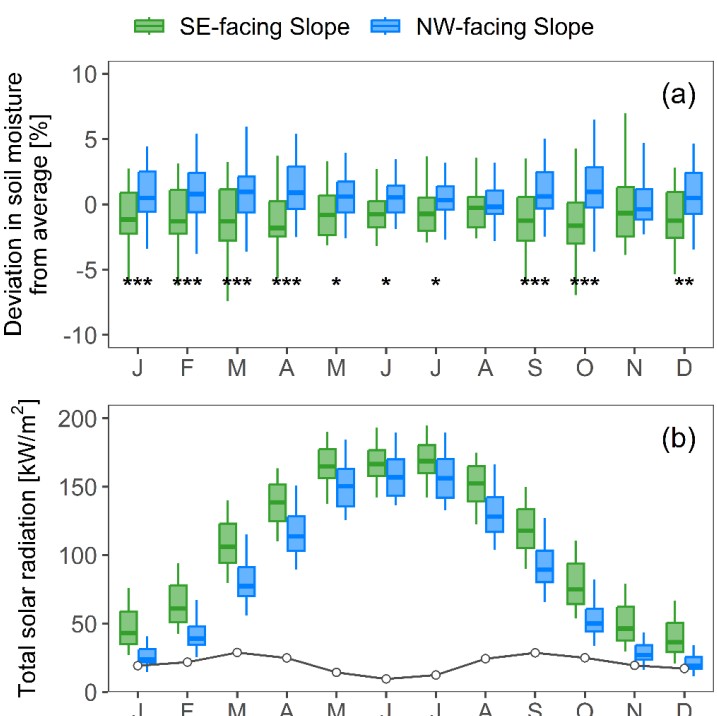

**Fig. 9. Seasonal patterns in volumetric soil moisture differences in the top 100 cm between the**
**SE-facing and NW-facing slopes ($\delta'\theta_{ij0-100}$) (a) and solar radiation reaching the two slopes (b).**
**In (a), \*, \*\*, and \*\*\* denote statistically significant differences at $\alpha = 0.05$, 0.01, and 0.001 levels,**
**respectively, determined by one-way ANOVA. The black line in (b) indicates the seasonal trend**
**of the difference in the total solar radiation for the two slopes. Differences in solar radiation and**
**soil moisture between the two slopes are smaller during the summer than during the rest of the**
**year.**

Some previous studies (e.g., Western et al. (2003) have reported that soil moisture patterns are

predominantly shaped by topographic convergence, and that these effects are stronger during the wet



season. By contrast, the soil moisture pattern on the slopes at our catchment was primarily shaped by

aspect, and was persistent during both wet and dry conditions (Fig. 7). We found no statistically

significant correlation between TWI and the soil moisture patterns on the slopes for any soil depth, or

averaged over the top 100 cm of the soils. Soil moisture is, however, markedly higher in the gully (Figs.

2-3), consistent with the high TWI values there. Topographic effects on soil moisture patterns are

typically mediated by lateral flow (Grayson and Western, 2001), but such flows are unlikely to be

dominant at the Gutun catchment, in view of the absence of impermeable bedrock or confining layers

in the thick and homogeneous loess deposits. Therefore, as a typical proxy of topography, TWI is

probably not a suitable index for explaining the soil moisture pattern in such systems (Dymond et al.,

2021).

### 4.3 Annual soil moisture storage change


Spatial patterns of annual soil moisture storage change (ΔS) at depths of 0-100 cm, 100-200 cm, and

200-300 cm are illustrated in Fig. 10. ΔS in the top meter of the soil exhibited a clear spatial pattern,

with the highest ΔS on the NW-facing slope, followed by intermediate ΔS on the SE-facing slope and

the lowest ΔS in the gully. In the 100-200 cm and 200-300 cm soils, ΔS was much smaller and also

more similar between the NW- and SE- facing slopes. This suggests that during the growing season,

more water was removed from root-zone soils on the NW-facing slope than on the SE-facing slope.

When considered together with the spatial pattern of soil moisture content in the 0-100 cm soils (Figs.

7 and 9a), these results suggest that the NW-facing slope contained more water than the SE-facing

slope during the dormant season, then lost more water during the growing season, but remained wetter

than the SE-facing slope at the end of the growing season.

These findings are consistent with the observations of Tromp-van Meerveld and McDonnell (2006) on

a hillslope of the Panola Mountain Research Watershed, Georgia, USA. In their midslope locations,

which had comparatively deep soils and high soil moisture storage, plants could obtain more water

from the soils without limiting transpiration in the growing season. In contrast, the upslope had lower

soil moisture storage, and reductions in soil moisture during the growing season restricted transpiration,

resulting in less water being extracted from these soils. At our site, differences in moisture storage arise

from energy-driven differences in ET rather than soil depth variations. Nonetheless, the differences in





ΔS between the NW-facing and SE-facing slopes in our study are consistent with the observations of

Tromp-van Meerveld and McDonnell, suggesting that the denser vegetation on the NW-facing slope

(Fig. 1b) may consume more water, thereby narrowing the soil moisture gap between the two slopes

during the growing season (Fig. 9a).

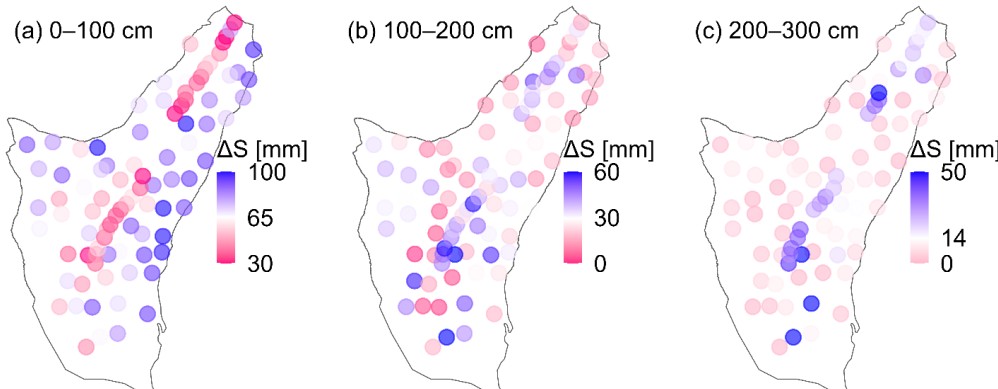

**Fig. 10. Spatial pattern of annual soil moisture storage change (ΔS) integrated over depths of 0-**
**100 cm (a), 100-200 cm (b), and 200-300 cm (c). In the top 100 cm of the soil profile (a), soil**
**moisture changes are much greater on the slopes than in the gully, and greater on the NW-facing**
**slope than on the SE-facing slope. These differences become less distinct between 100 and 200**
**cm (b), and the slopes become almost indistinguishable below 200 cm (c), with the greatest soil**
**moisture changes occurring in the gully instead. Note that the color scales differ among the three**
**panels. Blue colors indicate changes larger than the catchment average; red values indicate**
**changes that are smaller than the catchment average.**

Table 1 shows an estimated water balance for the Gutun catchment. During the soil moisture

monitoring periods, the annual average P was 556 mm/year, including 475 mm that fell between June

and October (the driest and wettest months, respectively; Fig. 6). The Penman-Monteith equation

yields an annual PET of 780 mm/year, including 420 mm between June and October. We estimated

groundwater recharge (G) at 45 mm/year based on streamflow measured at the outlet of a 24.3 km²

catchment that encompasses our field site; we chose this larger catchment because it is more likely to

capture baseflow, including the groundwater fluxes that might flow below the gauge in the small Gutun

headwater study area. Subtracting this value from annual precipitation yields an estimated 511

mm/year in actual evapotranspiration (AET), which is 66% of annual PET. Scaling the June-October

PET by this same factor yields an estimated 275 mm of AET between June and October. The water





balance between June and October can then be estimated as P minus the sum of AET and G, yielding

an expected soil moisture storage change of about 155 mm. The average seasonal ΔS that we measured in the 0-300 cm soils for the entire catchment was 110 mm. This is broadly similar to the water balance estimate, suggesting that the 0-300 cm soils account for most or all of the seasonal water storage in the Gutun catchment.

**Table 1 Estimated annual water balance of the Gutun catchment.**

| Items | Year [mm/year] | June-October [mm/season] |
|---|---|---|
| P | 556 | 475 |
| PET | 780 | 420 |
| G* | 45 | 45 |
| AET** | 511 | 275 |
| AET/PET** | 0.66 | 0.66 |
| P-AET-G** | 0 | 155 |
| ΔS in 0-300 cm | - | 110 |

Notes: * indicates estimate from measured streamflow; ** indicates estimate from annual water balance.

**5. Conclusion**

This study has documented the spatial patterns and seasonal dynamics of volumetric soil moisture in a small Loess Plateau catchment, using long-term measurements in a dense network of monitoring sites. The largest seasonal changes in soil moisture occurred in the upper 100 cm of the soils, with little change occurring below 260 cm. Within the upper 100 cm, soil moisture varied seasonally between wet and dry conditions, primarily due to the seasonal imbalance between PET and P. An aspect-

dependent spatial pattern in soil moisture on the hillslopes was particularly evident under wet conditions (but also observable under dry conditions), with the NW-facing slope exhibiting higher soil moisture than the SE-facing slope. The NW-facing slope also exhibited larger seasonal variations in soil moisture storage. Because the soil texture was uniform and there was no correlation between soil moisture and TWI across the slopes, variations in evapotranspiration appear to have controlled the

spatial pattern of hillslope soil moisture in the top 0-100 cm of the soil under both wet and dry

conditions. Water balance considerations also suggest that storage in the upper 300 cm of the soils

accounts for most or all of the seasonal water storage in the catchment. These observations contribute

to understanding runoff generation mechanisms in Loess Plateau catchments. They may also be useful

as reference values for sites with similar loess soils and highly seasonal climates.


**Data availability**

The dataset underlying our findings will be archived in an online repository and the DOI will be

included in the final published version of the paper.

**Acknowledgments**

This study was partly supported by the National Natural Science Foundation of China (Nos. 41971045

and 42107340), and the "Western Light" Innovation Cross Team Program, Chinese Academy of

Sciences. We thank the National Observation and Research Station of Earth Critical Zone on the Loess

Plateau for supporting our field work. We thank the Chinese Scholarship Council (No. 202106040099)

for financially supporting our Swiss-Chinese collaboration.

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
