# Peer review of "Seasonal dynamics and spatial patterns of soil moisture in a loess catchment"

_Hydrology and Earth System Sciences, 2023_

## Author Comment (AC1)

**We thank the anonymous reviewer for reviewing our manuscript and for providing helpful comments. Below we respond (in bold type) to the reviewer's comments (in normal type).**

In this study, intensive measurements of soil water content from 5 m profile in a watershed were made during 5 years. Spatial pattern and temporal dynamics of soil water content have been explored. The authors found some interesting results and further highlighted that evapotranspiration is the dominant mechanism of water flow under both wet and dry conditions on the Chinese Loess Plateau. The paper is well written and easy to understand. I think it worth publication in HESS. However, I think authors can further improve the expression of some equations and some other minor issues need to be addressed before it can be accepted.

**Thanks very much for these comments.**

Lines 17-18: this is too general and may not be true. Please be more specific regarding the knowledge gap please.

**We will rewrite the knowledge gap: _"The spatial and seasonal patterns in soil moisture and the processes controlling them in loess landscapes are not well understood."_**

Line 45-49: This was also observed in the study area before at the transect scale for different land uses. In addition, the associated patterns will be depending on what indicator (SD or CV) is used for characterizing spatial variability (https://doi.org/10.1016/j.geoderma.2011.02.008).

**We will include this reference: _"The spatial heterogeneity of soil moisture usually varies with the average field-, hillslope-, transect-, and catchment-scale wetness (Hu et al., 2011; Western et al., 2003)."_**

Line 56-57: spatial pattern of soil moisture was found to be dominated by topography in a watershed on the Chinese Loess Plateau (https://doi.org/10.1016/j.jhydrol.2013.10.002). I would encourage authors to discuss dynamics and mechanisms of soil moisture by drawing a bit more literature from the study area.

**The linked reference concluded that topographic properties, particularly the convergence index, significantly influenced soil moisture patterns in a catchment located in Saskatchewan, Canada, but played a negligible role in a catchment located on the Chinese Loess Plateau. The authors attribute this to the limited impact of lateral flow and the dominant role of vertical water movement in the semi-arid loess catchment.**

**We will include this reference in Introduction section (P3, Line 80): _"…even at the end of the growing season when the catchment was at its driest state. By contrast, in a semi-arid catchment_**

*in the Loess Plateau, China (annual precipitation 437 mm), Hu and Si (2014) reported that the convergence index had negligible impact on soil moisture patterns in both wet and dry conditions.*"

Line 76 and Line 95: please see my comments above. The most relevant literature on the Chinese Loess Plateau should be discussed for yielding knowledge gaps.

**We will discuss more studies conducted in the Chinese Loess Plateau and rewrite the knowledge gaps: "…that may result in distinct soil moisture patterns.** *Several studies have examined the spatial variability of soil moisture and its complex links with potential controlling factors on the Loess Plateau (Gao et al., 2011; Gao et al., 2016; Qiu et al., 2001; Yu et al., 2018). However, the relatively sparse observation sites and short monitoring period, combined with the highly seasonal local climate, make the spatial and seasonal patterns difficult to detect. Furthermore, the selection of the controlling factors can sometimes be subjective (Hu et al., 2017), with the result that we lack a systematic assessment of local and nonlocal controls on soil moisture patterns in this region.*"

Line 134: Authors please comment why 1 mm rather than 2 mm was used for soil texture analysis.

**In loess-based soils, the fraction of "very coarse sand" (with diameter limits from 1 mm to 2 mm) is generally very small or even negligible. This is supported by Wang et al. (2013), who reported that only a small number of sand particles were retained on the 1 mm mesh during soil sieving. Considering this, we made an adjustment to our soil texture measurement by using 1 mm soil sieving, which is a widely accepted practice in various studies conducted in the loess plateau. For instance, Hu et al. (2011), Jia et al. (2013), and Wang et al. (2011) have also adopted the same approach in their studies.**

Line 155: too many abbreviations to remember, how about using s, m, d and y, respectively, to represent site, month, depth and year?

**We appreciate the reviewer's suggestion to use more straightforward abbreviations. While these abbreviations s, m, d, and y can sometimes have multiple meanings, such as "second", "month", "day", and "year", respectively. This could lead to confusion, particularly when they are used in combination, for example, $\theta_{smd}$. To avoid such ambiguity, we decide to continue using i, j, and k as abbreviations for "site", "month", and "soil depth". The primary purpose of using mathematical notation with abbreviations is to aid readers in understanding the calculation process, while in the results and discussion sections, we will make sure to provide full names of variables alongside the abbreviations to clarify the variables being discussed. For example, in Figure 2, we will provide the caption as "*… the average volumetric soil moisture for hillslope ($\theta_{hillslope,jk}$) and gully sites ($\theta_{gully,jk}$)*". Similarly, in Figure 6, we will provide the caption as "*… average volumetric soil moisture in the top 100 cm of soil for SE-facing slope ($\theta_{SEf,j0-100}$), NW-facing slope ($\theta_{NWf,j0-100}$), and gully sites ($\theta_{gully,j0-100}$)*". Readers do not need to continuously remember which variable is represented by which mathematical notation, as the full names of the variables are provided for reference.**

Line 161-169: I would like to consider to embody soil moisture of different sub-regions or the whole area of the watershed in equation (2) by introducing another variable. The same may apply to eq. (3). As it is, the equations are not mathematically robust enough although they are understandable. You need to explain why you are specifically interested in 0-100 cm, does it have anything to do with your finding that soil moisture at 0-100 cm is more temporally dynamic?

**In equation (2) and equation (3), we will introduce "$\theta_{gully}$", "$\theta_{hillslope}$", "$\theta_{NWf}$", and "$\theta_{SEf}$" to represent the soil moisture for gully, hillslope, NW-facing slope, and SE-facing slope sites, respectively:**
*"Because of the much higher soil moisture in the gully than on the hillslopes, in each month $j$ and soil depth $k$, we also determined average soil moisture for all gully ($\theta_{gully,jk}$) and hillslope sites($\theta_{hillslope,jk}$), and for the NW- ($\theta_{NWf,jk}$) and SE-facing slopes sites ($\theta_{SEf,jk}$) separately:*

$$\theta_{location,jk} = \frac{1}{N} \sum_{i=1}^{N} \theta_{ijk} \quad , \tag{2}$$

*where $N$ is the number of gully (N=25), hillslope (N=64), NW-facing slope (N=30), or SE-facing slope (N=34) sites. We also determined the average soil moisture over 0-100 cm depth (5 soil layers) for the gully ($\theta_{gully,j0-100}$), NW-facing slope ($\theta_{NWf,j0-100}$), and SE-facing slope ($\theta_{SEf,j0-100}$) in each month $j$:*

$$\theta_{location,j0-100} = \frac{1}{5} \sum_{k=20}^{100} \theta_{location,jk} , \tag{3}$$

*"*

**We are more interested in soil moisture over 0-100 cm depth indeed because there are more pronounced seasonal changes in the soil moisture within these soil layers. We will clarify this after equation (3): *"We specifically focused on soil moisture in the top 100 cm, as our analysis in seasonal variability (see section 4.1) indicated that soil moisture within this depth range exhibits more pronounced seasonal dynamics."***

Line 179: it is a bit confusing as k refers to soil depth above (e.g., 20 cm, 40 cm, …), but here it represents the rank of the layer from top (e.g., 1, 2, …). Please be sure they are referring to the exact same thing. Can you use eq. (4) to calculate? In my mind you just need to make k more flexible, it can be a certain depth or a certain layer. The same to Eq (6), can you please use one equation to explain it clearly? To me, the main difference will be whether the whole watershed or just part of the watershed are involved in calculation. So please try to use as small number of equations as possible.

**We will unify "k" to refer to soil depth (e.g., 20 cm, 40 cm, …) consistently throughout the manuscript. The relevant equations (equations (3), (5), and (8)) will be adjusted to employ the notation "$\frac{1}{5}\sum_{k=20}^{100}$ ".**

We will merge equations (5) and (6) into a single equation to minimize the number of equations used: *"…Then we similarly determined the average seasonal deviation in soil moisture over 0-100 cm deep soils for the gully ($\delta\theta_{gully,j0-100}$), NW-facing slope ($\delta\theta_{NWf,j0-100}$), and SE-facing slope ($\delta\theta_{SEf,j0-100}$) separately in each month  j:*

$$\delta\theta_{location,j0-100} = \frac{1}{N}\sum_{i=1}^{N}(\frac{1}{5}\sum_{k=20}^{100}\delta\theta_{ijk}) \quad , \tag{5}$$

*where  N  is the number of gully (N=25), NW-facing slope (N=30), and SE-facing slope (N=34) sites."*

Line186-192: why these needed to be calculated, how does this relate to the main objective you want to target? They need to be better clarified.

The trimmed standard deviation (we will change it to the "regular" standard deviation based on the comments from reviewer#2) of $\theta_{ijk}$ at each site and depth ($\sigma_{ik}$) was used to quantify the seasonal changes in soil moisture. Then we identified the depth at which $\sigma_{ik}$ exhibited its maximum value or converged to a small value, allowing us to determine the depths at which the seasonal changes in soil moisture were largest or collapsed. In the revised manuscript, we will re-clarify this: *"We also quantified the seasonal changes in soil moisture at each site and depth ($\sigma_{ik}$) using the standard deviation (SD) of  $\theta_{ijk}$. We identified the depth of maximum  $\sigma_{ik}$  to determine the depth at which the seasonal changes in soil moisture were largest. We also identified the depth where  $\sigma_{ik}$  converges to a small value to determine the depth at which seasonal soil moisture changes collapse. We defined the collapse threshold as the minimum  $\sigma_{ik}$  plus 10% of the difference between the maximum and minimum  $\sigma_{ik}$  at each site. The shallowest depth at which $\sigma_{ik}$  was less than this threshold was defined as the depth at which the seasonal changes collapse."*

Similarly, In Line 205-207, the trimmed standard deviation (which also will be changed to "regular" standard deviation) of  $\theta_{ijk}$  at each month and depth ($\sigma_{jk}$) was used to quantify the overall spatial variability in soil moisture across the hillslopes. Then we used  $\sigma_{jk}$  and  $\theta_{hillslope,jk}$  to explore the relationship between spatial variability and soil moisture conditions across the hillslopes. We will add more clarification in the revised manuscript: *"The spatial variability of the soil moisture in month  j  and soil layer k, across the hillslope,  $\sigma_{jk}$, was described by the SD of  $\theta_{ijk}$. We used  $\sigma_{jk}$ and  $\theta_{hillslope,jk}$  to explore the relationship between spatial variability and soil moisture conditions across the hillslopes."*

Line 198 and Line 203: you need try to find a way to embody slope in the equation as I commented above. Why $\delta'$rather than $\delta$ is used here?

We will incorporate  $\theta_{hillslope,jk}$  in this equation, which has been calculated in equation (2) as recommend by the reviewer: *"…thus the spatial deviation in soil moisture for hillslope site  i, soil depth  k, and month  j,  $\delta'\theta_{ijk}$  was computed as*

$$\delta'\theta_{ijk} = \theta_{ijk} - \theta_{hillslope,jk} \quad , \tag{6}$$

*where  $\theta_{hillslope,jk}$  is the average soil moisture for the hillslope sites in month  j  and soil depth  k,*

*as described above."*

$\delta$ was used in the section "3.2.1. Seasonal variability in soil moisture" to represent the "seasonal deviation" in soil moisture. For instance, $\delta\theta_{ijk}$ indicates the deviation in the soil moisture $\theta_{ijk}$ from the annual average for that site and depth. Therefore, $\delta'$ was introduced in the section "3.2.2 Spatial variability in soil moisture at hillslope scale" to represent the "spatial deviation" in soil moisture. For instance, $\delta'\theta_{ijk}$ indicates the deviation in the hillslope soil moisture $\theta_{ijk}$ from the hillslope average for that month and depth.

Line 224: why not other topographic properties such as slope?

**Thanks for this suggestion. We have tried this in the data analysis process, but the soil moisture and slope are not correlated well in our study area. Here is the Spearman correlation result:**

| Month | Correlation coefficient | p-value | p-range |
|---|---|---|---|
| 1 | -0.194028418 | 0.127403437 | p>0.05 |
| 2 | -0.189708141 | 0.136225605 | p>0.05 |
| 3 | -0.174299155 | 0.171477654 | p>0.05 |
| 4 | -0.169402842 | 0.183973073 | p>0.05 |
| 5 | -0.119719662 | 0.349104022 | p>0.05 |
| 6 | -0.049731183 | 0.698038435 | p>0.05 |
| 7 | -0.124663978 | 0.329446406 | p>0.05 |
| 8 | -0.020593318 | 0.872499752 | p>0.05 |
| 9 | -0.18327573 | 0.150207591 | p>0.05 |
| 10 | -0.156586022 | 0.219785145 | p>0.05 |
| 11 | -0.197148618 | 0.121308046 | p>0.05 |
| 12 | -0.179723502 | 0.158375369 | p>0.05 |

Line 232: why not try cos(aspect) as it looks like a good indicator for soil moisture (https://doi.org/10.1016/j.jhydrol.2017.05.054)

**Thanks for this suggestion. We have tried the correlation analysis between soil moisture and cos(aspect), but they are not correlated well in our study area. Here is the Spearman correlation result:**

| Month | Correlation coefficient | p-value | p-range |
|---|---|---|---|
| 1 | 0.028705837 | 0.822916352 | p>0.05 |
| 2 | 0.023521505 | 0.854535517 | p>0.05 |
| 3 | -0.021505376 | 0.866896968 | p>0.05 |
| 4 | -0.130760369 | 0.306197794 | p>0.05 |
| 5 | -0.012480799 | 0.922579921 | p>0.05 |
| 6 | 0.039458525 | 0.758281893 | p>0.05 |
| 7 | 0.018097158 | 0.88786502 | p>0.05 |

| 8 | 0.12562404 | 0.325712633 | p>0.05 |
|---|---|---|---|
| 9 | 0.002208141 | 0.98640505 | p>0.05 |
| 10 | 0.084149386 | 0.511067736 | p>0.05 |
| 11 | -0.178379416 | 0.161550421 | p>0.05 |
| 12 | 0.029473886 | 0.818254664 | p>0.05 |

**Instead a correlation test, we linked the difference in monthly soil moisture and the difference in incoming solar radiation between the NW-facing and SE-facing slopes to determine the effect of aspect on soil moisture (Figs. 7 and 9).**

Figure 6c: title from y axis is missing. Can you please add measurement error for each graph?

**We will add a title for the y-axis as "*P, PET, and P-PET [mm/month]*". Error bars will also be added on each panel. We will also fix a few minor errors for the monthly P and PET calculation, resulting in slight discrepancies for the result. It is important to note that this correction does not affect our conclusions.**

Line340-342: this is not that obvious visually. I would suggest authors to improve figure 7 by also making sure they are readable in white and black.

**If only white and black colors are used to represent dry and wet conditions, we would miss capturing the seasonal changes in spatial variation (smaller spatial variation from May to July, larger spatial variation from October to March, which are now depicted with lighter and darker colors, respectively). What we can do is add an aspect layer as a base map to help support the findings.**

Line 350-356: I don't think the convex-upward model is necessarily applicable to Chinese Loess Plateau. Even mean soil water content higher than 20% was measured, this pattern was not observed in a previous study (https://doi.org/10.1016/j.geoderma.2011.02.008). Authors may want to discuss a bit more here.

**We will include this reference and rewrite the discussion as follows: "*The most widely reported model for describing the relation between spatial heterogeneity and mean soil moisture is a convex-upward parabola, with spatial variability peaking at intermediate values of soil moisture content (approximately 20%) (Brocca et al., 2010; Famiglietti et al., 2008; Jarecke et al., 2021; Peterson et al., 2019; Tague et al., 2010; Western et al., 2003). This convex parabola has been observed in loess catchments as well (Gao et al., 2011; Gao et al., 2015; Shi et al., 2014), where spatial variability peaked at soil moisture within 15%-20%. However, in a similar loess system, Hu et al. (2011) found that the spatial variability slightly increased with increasing soil moisture, even in wet conditions (20%-25%), indicating that a natural logarithmic curve might better describe the relationship between spatial variability and average soil moisture. In the Gutun catchment, the average soil*"**

*moisture mainly concentrated between 5%-15%, which means we may only be observing the short rising segment of a convex parabola below the variability peak, or the middle section of a logarithmic curve.*"

Line 373-374: again, this is not that obvious though this may be true. Please consider to improve the figure 7. Probably rather than showing point value, it would be good idea to mapping the whole area?

**Although certain methods, such as Kriging interpolation, can map the soil moisture distribution for the entire catchment, they may generate artificial values, including strange ones in areas lacking monitoring sites. This would be misleading. Instead we prefer to present the actual measured values, which we believe are more reliable and informative. We will add an aspect layer as a base map into Fig. 7 to help illustrate the findings.**

Line 405-406: the statistical analysis here can be misleading as less samples from gully than slope.

**The correlation analysis was specifically based on monitoring data from the hillslope sites, excluding the gully areas. Therefore, our result indicates that there is no significant correlation between TWI and soil moisture patterns at the hillslope scale. We will clarify this point and highlight the "hillslope scale" in the revised manuscript: "*…We found no statistically significant correlation between TWI and the soil moisture patterns on the slopes for any soil depth, or averaged over the top 100 cm of the soils in each month. It is important to emphasize that we focus on the relationship between TWI or aspect with soil moisture patterns at the hillslope scale, excluding the gully. Soil moisture at the catchment scale is, however, markedly higher in the gully (Figs. 2-3), consistent with the high TWI values there.*"**

Line 411-413: I would be cautious to draw this conclusion because gully was much wetter in this study.

**We will emphasize the "hillslope scale" in the conclusion: "*Therefore, as a typical proxy of topography, TWI is probably not a suitable index for explaining the soil moisture pattern on the hillslopes in such systems (Dymond et al., 2021).*"**

Line 426: the role of aspect in driving water variation was also documented in previous studies in the same areas. These papers need to be included in discussion.

**We will include several related studies conducted in the loess plateau in L383: "*…leading to smaller differences in evaporation and thus more consistent soil moisture between the two slopes at the Gutun catchment. Hu et al. (2017) and Gao et al. (2016) similarly showed a pronounced impact of aspect on soil moisture patterns in other catchments in loess landscapes similar to ours.*"**

References:

*Hu, W., Shao, M., Han, F. and Reichardt, K., 2011. Spatio-temporal variability behavior of land surface soil water content in shrub-and grass-land. Geoderma, 162: 260-272.*

*Jia, X., Shao, M., Wei, X. and Wang, Y., 2013. Hillslope scale temporal stability of soil water storage in diverse soil layers. Journal of Hydrology, 498: 254-264.*

*Wang, Y., Shao, M., Zhu, Y. and Liu, Z., 2011. Impacts of land use and plant characteristics on dried soil layers in different climatic regions on the Loess Plateau of China. Agricultural and Forest Meteorology, 151: 437–448.*

*Wang, Y., Shao, M., Liu, Z. and Horton, R., 2013. Regional-scale variation and distribution patterns of soil saturated hydraulic conductivities in surface and subsurface layers in the loessial soils of China. Journal of Hydrology, 487: 13–23.*

---

## Author Comment (AC2)

**We thank Kendra Kaiser for reviewing our manuscript and for providing helpful comments. Below we respond (in bold type) to the reviewer's comments (in normal type).**

This study synthesizes information collected from a dense soil moisture monitoring network in a watershed with deep loess soils. They have highlighted the limitations of using the topographic wetness index for understanding the spatial variability of soil moisture in places that do not have a shallow confining layer, and that the spatial variability in these hillslopes is largely driven by evapotranspiration. The introduction is well written, while the discussion could use some editing to highlight the main findings and the more clearly share the associated story. Below in my specific comments I give suggestions on how to edit and trim the discussion to help focus on the most interesting findings.

**Thanks very much for these comments.**

L23 – confusing since only part of the year is captured in these distinctions

**Based on the seasonal patterns in soil moisture (Figs. 6a and 6b), it is evident that May to July and October to March were under dry and wet conditions, respectively. April was defined as the wet-to-dry transition month, and August and September were defined as the dry-to-wet transition months (L314-315), because their deviation from annual average are around 0 (Fig. 6b). These three months bridge the wet and dry conditions and it is challenging to precisely assign them to a specific wet or dry condition.**

L114 – Gully land consolidation --- I'm not sure if this is necessary here, but I am curious why this is being done and if it is occurring in other watersheds in the area.

**The Gully land consolidation is a project undertaken by the Chinese government in the Chinese Loess Plateau with an investment of approximately 4.6 billion dollars, aimed to address the problem of farmland losses caused by the local reforestation activities. Starting in 2000, the Chinese government undertook a large-scale reforestation effort to address the severe erosion problem in the loess plateau. In this process, certain agricultural lands in the hillslope were abandoned and dedicated to grass/tree planting or natural vegetation recovery. To prevent the adverse impact of farmland abandonment on local farmers' income, the gully land consolidation project was implemented alongside reforestation: filling and levelling the gullies with soils from adjacent hillslopes, converting them into new farmland suitable for crop cultivation. As a consequence, the topography of the catchments has been altered compared to natural undisturbed ones, especially in the gully areas, making them into a newly typical and widespread catchment type in the loess plateau. Although our study was not primarily focused on the hydrological changes resulting from this project, we think it is necessary to acknowledge this point to distinguish such a disturbed catchment from natural ones.**

L187 – why the 20% trimmed standard deviation?

**To avoid such similar confusion for readers, we will use non-trimmed standard deviation to calculate the temporal and spatial variabilities. Re-calculation using non-trimmed standard deviation gives similar results to our robust estimates of variability (20% trimmed standard deviations used now), thus it does not alter our conclusions. We will update the new results in Fig. 4 and Fig. 8.**

L238 & 305 – suggest to edit sentences as to not start with "Fig. 2/6"

**We will edit the sentence in L238: "*The seasonal changes in the average soil moisture for each depth for the hillslope ($\theta_{hillslope,jk}$) and the gully ($\theta_{gully,jk}$), respectively, are shown in Fig. 2.*"**

**We will edit the sentence in L305: "*The seasonal patterns in average soil moisture over the top 100 cm for the hillslope and gully sites ($\theta_{SEf,j0-100}$, $\theta_{NWf,j0-100}$, and $\theta_{gully,j0-100}$), and the deviations from their annual averages ($\delta\theta_{SEf,j0-100}$, $\delta\theta_{NWf,j0-100}$, and $\delta\theta_{gully,j0-100}$), are illustrated in Fig. 5.*"**

L290 – figure 4 caption is a little unclear, it seems as though the figure labels should be before the description, e.g. a) depth of max trimmed stdv, b) depth of collapse. How are 90% of sites "collapsing" between 20-100cm and 82% of sites collapsing between 160 and 260?

**We will relocate the figure labels to appear before the corresponding descriptions for Fig. 4, also for the other figures.**

**Fig. 4 displays the tallied histogram representing the frequency distribution in depth of maximum trimmed standard deviation (TSD) and in depth at which TSD collapses for 72 monitoring sites. The x-axis indicates the count of the monitoring sites. In panel (a), we observed that the maximum TSD occurred in the depth of 0-20 cm for 15 monitoring sites, in the depth of 20-40 cm for 20 sites, 40-60 cm for 8 sites, 60-80 cm for 17 sites, and 80-100 cm for 5 sites. Therefore, a total of 65 sites (=15+20+8+17+5) showed the maximum SD within the top 100 cm soil depth. As a percentage of the total number of sites (excluding those were measurements are truncated before 5m depth), this is 65/72 ≈ 90%. Therefore, we concluded that the depth of the maximum SD was between 0 and 100 cm for 90% of the sites. By the same way, we concluded that the depth at which the SD collapses was between 160 and 260 cm for 82% of the sites. We will change the x-axis title from "*Frequency*" to "*Count of sites*" to enhance the clarity. The percentage number will be a bit different after the result updated (re-calculation by non-trimmed SD), but this will not affect our conclusions.**

L282-286 is a longer description of the content in L 316-321 which is easier to read and is associated with a mechanism, consider removing the former.

**We will remove the content in L282-286. Additionally, we will exchange the order of Fig. 5 and Fig. 6, then include a simple sentence describing the tallied histograms to complement the main finding in L316-321:** *"…, with wet-to-dry transitions around April and dry-to-wet transitions between August and September (Figs. 5a-b). Together with the tallied histograms of the months in which the annual maximum and minimum soil moisture occurred for each soil layer (Fig. 6), these results suggest that soil moisture in the top 100 cm of soil was at a minimum in the late spring and early summer, …"*

L298 – What is the color scheme in Fig 5 for?

**We will remove the color scheme and fill it with single color for Fig. 5a and 5b, respectively.**

Fig 6 could be easier to read as a vertically stacked figure so that months are aligned

**We will re-stack the panels in a vertical way.**

L341 – figure color descriptions should be in a legend or in caption, not in the text

**We will remove figure color descriptions from L341 and add them in the caption of Fig. 7**: *"Deviation in volumetric soil moisture (0-100 cm, $\delta'\theta_{ij0-100}$) from monthly average across hillslope sites. The lighter colors (deviation close to 0) indicate smaller deviations from the average soil moisture, while darker colors indicate larger deviations. Blue and red colors indicate soil moisture above and below the average, respectively."*

L 349 – this is slightly misleading as written given that the max soil moisture is below 20% in the hillslopes. You get there in the paragraph, but the set up suggests that there is something different happening here than in the literature.

**We will revise this paragraph considering the comments from you and reviewer #1:** *"The most widely reported model for describing the relation between spatial heterogeneity and mean soil moisture is a convex-upward parabola, with spatial variability peaking at intermediate values of soil moisture content (approximately 20%) (Brocca et al., 2010; Famiglietti et al., 2008; Jarecke et al., 2021; Peterson et al., 2019; Tague et al., 2010; Western et al., 2003). This convex parabola has been observed in loess catchments as well (Gao et al., 2011; Gao et al., 2015; Shi et al., 2014), where spatial variability peaked at soil moisture within 15%-20%. However, in a similar loess system, Hu et al. (2011) found that the spatial variability slightly increased with increasing soil moisture, even in wet conditions (20%-25%), indicating that a natural logarithmic curve might better describe the relationship between spatial variability and average soil moisture. In the Gutun catchment, the average soil moisture mainly concentrated between 5%-15%, which means we may only be observing the short rising segment of a convex parabola below the variability peak, or the middle*

*section of a logarithmic curve.”*

Figure 7 –an aspect layer as a basemap to help support the findings re: aspect and moisture patterns would be helpful; label the legend and edit the caption to start with "Deviation in VWC from ...

**Thanks for this suggestion. We will add an aspect layer as a basemap in Fig. 7.**
**The legend will be labelled with "*Deviation in soil moisture*".**
**We will edit the caption: *"Deviations in volumetric soil moisture (0-100 cm, $\delta' \theta_{ij0-100}$) from monthly averages across hillslope sites."***

L374 – is this statement supported given the high amount of overlap between boxplots in Fig9? Is it statistically significant? There can be a pattern without being statistically significant, but it should be clarified.
**We have tested the significance of difference with one-way ANOVA, the result is as follows:**

| Month | p-value | p-range | note |
|---|---|---|---|
| 1 | 0.000715 | p<0.001 | *** |
| 2 | 0.000773 | p<0.001 | *** |
| 3 | 0.000912 | p<0.001 | *** |
| 4 | 0.00000863 | p<0.001 | *** |
| 5 | 0.0213 | p<0.05 | * |
| 6 | 0.0191 | p<0.05 | * |
| 7 | 0.0297 | p<0.05 | * |
| 8 | 0.626 | ns | ns |
| 9 | 0.0000535 | p<0.001 | *** |
| 10 | 0.0000915 | p<0.001 | *** |
| 11 | 0.703 | ns | ns |
| 12 | 0.00180 | p<0.01 | ** |

**For most of the months (not August and November), the differences in soil moisture between the two slopes were statistically significant (under dry conditions, May to July, the difference is less distinct). We have noted the result for significance test in the Fig. 9(a) and its caption. We will further note it in L374 in the revised manuscript: "*Fig. 7 also reveals a spatial pattern in soil moisture, with the NW-facing slope being much wetter than the SE-facing slope, which was also seen in Fig. 5a and is quantified in more detail, including significance test, in Fig. 9.*"**

L410-412 – this is the thing that I find most interesting and broadly relevant, but clarify this is referring to spatial variability in the hillslopes (not the hillslope vs gully)

**We will clarify this as suggested: *"We found no statistically significant correlation between TWI and the soil moisture patterns on the slopes for any soil depth, or averaged over the top 100 cm of the soils in each month. It is important to emphasize that we focus on the relationship between TWI or***

*aspect with soil moisture patterns at the hillslope scale, excluding the gully. Soil moisture at the catchment scale is, however, markedly higher in the gully (Figs. 2-3), consistent with the high TWI values there."*

Section 4.3 –consider editing these first two paragraphs to be more integrated, e.g. L422-425 reads like results while L435-437 gets into the context that makes them interesting.

**We will edit these two paragraphs to be more integrated.**

L450 – confusing as written – most of the precipitation fell between driest and wettest months?

**We will remove the entire paragraph in which the L450 is located as suggested by the next comment.**

L449 – this paragraph isn't particularly compelling as written, it walks through calculations and results presented in the table. Move the table to results and retain the last sentence in the discussion.

**We will remove this paragraph, and move the last sentence into the first paragraph of Section 4.3: "*Spatial patterns of annual soil moisture storage change (ΔS) at depths of 0-100 cm, 100-200 cm, and 200-300 cm are illustrated in Fig. 10. The average ΔS that we measured in the 0-300 cm soils for the entire catchment was 110 mm. This is broadly similar to the water balance estimate (Table 1), suggesting that the 0-300 cm soils account for most or all of the seasonal water storage in the Gutun catchment.*"**

**We will also fix a few minor errors in the annual water balance estimation resulting in slight discrepancies in the revised manuscript. This correction does not affect our conclusions.**

---

## Author Comment (AC3)

**We thank Yanhui Wang for reviewing our manuscript and for providing helpful comments. Below we respond (in bold type) to the reviewer's comments (in normal type).**

This study explored the seasonal dynamics and spatial patterns of soil moisture by intensive measurements of soil water content from 5 m profile in a loess catchment in 2016-2021. The results have practical implications for catchment-scale hydrologic modeling and the design of soil moisture monitoring networks. The paper is well written, but there are a few minor issues that need improvement before acceptance.

**Thank you very much for the positive evaluation.**

Line 18: Isn't it 5.5 years from April 2016 to October 2021?

**We will correct 6.5 years to 5.5 years: "*In this study, volumetric soil moisture was monitored monthly for 5.5 years at 20 cm intervals between the surface and 500 cm depth at 89 sites across a small (0.43 km2) catchment on the Chinese Loess Plateau.*"**

Line 95-96: Please explain what local and nonlocal controls specifically denote.

**We have provided a comprehensive overview of the "local" and "non-local" controls in the first paragraph of the Introduction section, please see Lines 53-60: "*Grayson et al. (1997) and Western et al. (2003) demonstrated that topography has a greater influence on spatial patterns of soil moisture under wet conditions, due to redistribution of soil moisture by lateral flow, resulting in wetter soils along hillslope drainage lines in convergent topography. Under dry conditions, by contrast, soil properties and vegetation become more important factors because soil moisture is mainly affected by point-scale vertical water fluxes. Any topographic influence under dry conditions is more likely to be due to aspect rather than topographic convergence (Grayson and Western, 2001). Grayson and Western (2001) summarized this phenomenon as local and nonlocal control on soil moisture under dry and wet conditions, respectively.*"**

Line 260: The title from x axis is missing in Fig. 2., and the x axis scale capital letter meaning should also be stated. Check the other figures in the paper in the same way.

**We will add the note concerning the capital letters in the x-axis of Fig. 2: "*The capital letters in the x-axis indicate the months from January to December. The same abbreviations were used in the other figures.*" We believe the x-axis title can be omitted with the help of the new added notes.**

Line 262-264: The explanation of the data results should appear in the results and discussion rather than in the figure title. Other figure titles in the paper also have this problem, please modify it.

We believe that providing interpretations within the figure captions can improve the overall readability of the manuscript. Readers can directly grasp the main message conveyed by the figures without having to flip back and forth to the main text. Furthermore, many readers scan the figures of a paper without reading the entire text; putting the main points in the figure captions allows these readers to immediately grasp most of the main points of the paper. For these reasons, some academic journals require figures to have informative captions to enhance the overall quality of the manuscript. We will retain the explanatory content in figure captions, as part of a deliberate strategy that allows many readers to immediately understand what the figures mean, and to grasp many of the main points of the paper even if they do not read the text.

Line 402-404: "Some previous studies (e.g., Western et al. (2003) have reported that soil moisture patterns are predominantly shaped by topographic convergence, and that these effects are stronger during the wet season". The grammar of this sentence is wrong, please correct it.

**We will correct the grammar: "*Some previous studies (e.g., Western et al. (2003) have reported that soil moisture patterns are predominantly shaped by topographic convergence, and that this effect is stronger during the wet season.*"**

---

## Author Response (AR1)

**Point-by-point response to reviewer comments on "Seasonal dynamics and spatial patterns of soil moisture in a loess catchment" by Shaozhen Liu et al.**

**We would like to thank one anonymous reviewer, Kendra Kaiser, and Yanhui Wang for reviewing our manuscript and for providing helpful comments. The point-by-point reply to the comments is given below. The comments provided by the reviewers are shown in** normal font**, and our responses in bold.**

**1) Author's response to the comments by anonymous reviewer #1**

In this study, intensive measurements of soil water content from 5 m profile in a watershed were made during 5 years. Spatial pattern and temporal dynamics of soil water content have been explored. The authors found some interesting results and further highlighted that evapotranspiration is the dominant mechanism of water flow under both wet and dry conditions on the Chinese Loess Plateau. The paper is well written and easy to understand. I think it worth publication in HESS. However, I think authors can further improve the expression of some equations and some other minor issues need to be addressed before it can be accepted.

**Thank you very much for these positive comments. We have addressed all comments in detail below (in bold font).**

Lines 17-18: this is too general and may not be true. Please be more specific regarding the knowledge gap please.

**We rewrote the knowledge gap:** _**"The spatial and seasonal patterns in soil moisture and the processes controlling them in loess landscapes are not well understood."**_

Line 45-49: This was also observed in the study area before at the transect scale for different land uses. In addition, the associated patterns will be depending on what indicator (SD or CV) is used for characterizing spatial variability (https://doi.org/10.1016/j.geoderma.2011.02.008).

**We now include this reference:** _**"The spatial heterogeneity of soil moisture usually varies with the average field-, hillslope-, transect-, or catchment-scale wetness (Hu et al., 2011; Western et al., 2003)."**_

Line 56-57: spatial pattern of soil moisture was found to be dominated by topography in a watershed on the Chinese Loess Plateau (https://doi.org/10.1016/j.jhydrol.2013.10.002). I would encourage authors to discuss dynamics and mechanisms of soil moisture by drawing a bit more literature from the study area.

The study of the linked reference concluded that topographic properties, particularly the convergence index, significantly influenced soil moisture patterns in a catchment located in Saskatchewan, Canada, but played a negligible role in a catchment located on the Chinese Loess Plateau. The authors attribute this to the limited impact of lateral flow and the dominant role of vertical water movement in the semi-arid loess catchment. This is in agreement with our results. We include this reference in the Introduction section (P3, Line 80): "…*even at the end of the growing season when the catchment was at its driest state. By contrast, in a semi-arid catchment in the Loess Plateau, China (annual precipitation 437 mm, volumetric soil moisture <20%), Hu and Si (2014) reported that the convergence index had negligible impact on soil moisture patterns in both wet and dry conditions.*"

Line 76 and Line 95: please see my comments above. The most relevant literature on the Chinese Loess Plateau should be discussed for yielding knowledge gaps.

We now discuss more studies conducted in the Chinese Loess Plateau and have rewritten the knowledge gaps: "…*that may result in distinct soil moisture patterns. Several studies have examined the spatial variability of soil moisture and its complex links with potential controlling factors on the Loess Plateau (Gao et al., 2011; Gao et al., 2016; Qiu et al., 2001; Yu et al., 2018). However, the relatively low number of observation sites and short monitoring periods, combined with the highly seasonal local climate, make the spatial and seasonal patterns difficult to detect. Furthermore, the selection of the controlling factors can sometimes be subjective (Hu et al., 2017), with the result that we lack a systematic assessment of local and nonlocal controls on soil moisture patterns in this region.*"

Line 134: Authors please comment why 1 mm rather than 2 mm was used for soil texture analysis.

In loess-based soils, the fraction of "very coarse sand" (with diameters from 1 mm to 2 mm) is generally very small or even negligible. This is supported by Wang et al. (2013), who reported that only a small number of sand particles were retained on the 1 mm mesh during sieving. Considering this, we made an adjustment to our soil texture measurement by using 1 mm soil sieving, which is a widely accepted practice and used in various studies conducted in the loess plateau. For instance, Hu et al. (2011), Jia et al. (2013), and Wang et al. (2011) adopted the same approach in their studies.

Line 155: too many abbreviations to remember, how about using s, m, d and y, respectively, to represent site, month, depth and year?

We appreciate the reviewer's suggestion to use more straightforward abbreviations. While these abbreviations s, m, d, and y can sometimes have multiple meanings, such as "second", "month", "day", and "year", respectively. This could lead to confusion, particularly when they are used in combination, for example, $\theta_{smd}$. To avoid such ambiguity, we decided to continue using i, j, and k

as abbreviations for "site", "month", and "soil depth". The primary purpose of using the mathematical notation with abbreviations is to aid readers in understanding the calculation process (that is also described in words in this section). In the results and discussion sections, we now provide the full names of variables alongside the abbreviations to clarify the variables being discussed. For example, in Figure 2, we now provide the caption as *"… the average soil moisture for hillslope ($\theta_{hillslope,j,k}$) and gully sites ($\theta_{gully,j,k}$)"*. Similarly, in Figure 6, we rewrote the caption to "…*average soil moisture in the top 100 cm of soil for the SE-facing slope ($\theta_{SE,j,0-100}$), NW-facing slope ($\theta_{NW,j,0-100}$), and gully sites ($\theta_{gully,j,0-100}$)*". Thus, readers do not need to continuously remember which variable is represented by which mathematical notation, as the full names of the variables are provided for reference. We have furthermore double checked that we always use the same order (site, month, depth) when explaining things and added commas to the subscripts for improved readability.

Line 161-169: I would like to consider to embody soil moisture of different sub-regions or the whole area of the watershed in equation (2) by introducing another variable. The same may apply to eq. (3). As it is, the equations are not mathematically robust enough although they are understandable. You need to explain why you are specifically interested in 0-100 cm, does it have anything to do with your finding that soil moisture at 0-100 cm is more temporally dynamic?

In equation (2) and equation (3), we now introduce "$\theta_{gully}$", "$\theta_{hillslope}$", "$\theta_{NW}$", and "$\theta_{SE}$" to represent the soil moisture for gully, hillslope, NW-facing slope, and SE-facing slope sites, respectively:
*"Because of the much higher soil moisture in the gully than on the slopes, we also determined the average soil moisture for each month $j$ and soil depth $k$, for all gully ($\theta_{gully,j,k}$) and slope sites ($\theta_{hillslope,j,k}$), and for the sites on the NW- ($\theta_{NW,j,k}$) and SE-facing slopes ($\theta_{SE,j,k}$):*

$$\theta_{location,j,k} = \frac{1}{N}\sum_{i=1}^{N}\theta_{i,j,k} \quad , \tag{2}$$

*where $N$ is the number of gully (N=25), hillslope (N=64), NW-facing slope (N=30), or SE-facing slope (N=34) sites.*
*We also determined the average soil moisture over 0-100 cm depth (5 soil layers) for the gully ($\theta_{gully,j,0-100}$), NW-facing slope ($\theta_{NW,j,0-100}$), and SE-facing slope ($\theta_{SE,j,0-100}$) in each month $j$:*

$$\theta_{location,j,0-100} = \frac{1}{5}\sum_{k=20}^{100}\theta_{location,j,k} \quad , \tag{3}$$

*"*

Indeed, we are more interested in soil moisture over 0-100 cm depth because the seasonal changes in soil moisture are more pronounced for these soil layers. We now clarify this after equation (3):
*"We specifically focused on soil moisture in the top 100 cm, as our analysis of the seasonal variability in soil moisture (see section 4.1) indicated that soil moisture within this depth range exhibited more pronounced seasonal dynamics."*

Line 179: it is a bit confusing as k refers to soil depth above (e.g., 20 cm, 40 cm, …), but here it represents the rank of the layer from top (e.g., 1, 2, …). Please be sure they are referring to the exact same thing. Can you use eq. (4) to calculate? In my mind you just need to make k more flexible, it can be a certain depth or a certain layer. The same to Eq (6), can you please use one equation to explain it clearly? To me, the main difference will be whether the whole watershed or just part of the watershed are involved in calculation. So please try to use as small number of equations as possible.

**We unified the use of "k" to refer to soil depth (e.g., 20 cm, 40 cm, …) consistently throughout the manuscript. The relevant equations (equations (3), (5), and (8)) were adjusted and now employ the notation "$\frac{1}{5}\sum_{k=20}^{100}$ ".**

**We also merged equations (5) and (6) into a single equation to minimize the number of equations used: "_…Then we similarly determined the average seasonal deviation in soil moisture over 0-100 cm for the gully ($\delta\theta_{gully,j,0-100}$), NW-facing slope ($\delta\theta_{NW,j,0-100}$), and SE-facing slope ($\delta\theta_{SE,j,0-100}$) separately for each month j:_**

$$\delta\theta_{location,j,0-100} = \frac{1}{N}\sum_{i=1}^{N}(\frac{1}{5}\sum_{k=20}^{100}\delta\theta_{i,j,k}) \quad , \tag{5}$$

**_where N is the number of gully (N=25), NW-facing slope (N=30), and SE-facing slope (N=34) sites._"**

Line186-192: why these needed to be calculated, how does this relate to the main objective you want to target? They need to be better clarified.

**The trimmed standard deviation (we changed it to the "regular" standard deviation based on the comments from reviewer#2) of $\theta_{i,j,k}$ at each site and depth ($\sigma_{i,k}$) was used to quantify the seasonal changes in soil moisture. Then we identified the depth at which $\sigma_{i,k}$ was maximum or converged to a small value. This allowed us to determine the depths at which the seasonal changes in soil moisture were largest or collapsed. In the revised manuscript, we explain this better: "_We quantified the seasonal changes in soil moisture at each site and depth using the standard deviation (SD) of $\theta_{i,j,k}$ ($\sigma_{i,k}$). We identified the depth of the maximum $\sigma_{i,k}$ to determine the depth at which the seasonal changes in soil moisture were the largest. We also identified the depth where $\sigma_{i,k}$ converges to a small value to determine the depth below which seasonal soil moisture changes collapse (i.e., become very small). We defined a collapse threshold based on the minimum $\sigma_{i,k}$ plus 10% of the difference between the maximum and minimum $\sigma_{i,k}$ for each site. The shallowest depth at which $\sigma_{i,k}$ was less than this threshold was defined as the depth at which the seasonal changes collapses._"**

**Similarly, In Line 205-207, the trimmed standard deviation (which was also changed to the "regular" standard deviation) of $\theta_{i,j,k}$ for each month and depth ($\sigma_{j,k}$) was used to quantify the overall spatial variability in soil moisture across the hillslopes. Then we used $\sigma_{j,k}$ and $\theta_{hillslope,j,k}$ to explore the relationship between the spatial variability and soil moisture conditions across the**

hillslopes. We added more clarification in the revised manuscript: *"The overall spatial variability in soil moisture across the hillslopes, for each month and depth, was also quantified using the standard deviation. The spatial variability of soil moisture in month $j$ and soil layer $k$, across the hillslopes was described by the SD of $\theta_{i,j,k}$ ($\sigma_{j,k}$). We used $\sigma_{j,k}$ and $\theta_{hillslope,j,k}$ to explore the relationship between the spatial variability in soil moisture and the average soil moisture across the hillslopes."*

Line 198 and Line 203: you need try to find a way to embody slope in the equation as I commented above. Why $\delta'$ rather than $\delta$ is used here?

We incorporated $\theta_{hillslope,j,k}$ in this equation, which has been calculated in equation (2) as recommend by the reviewer: *"Thus, the spatial deviation in soil moisture for slope site $i$, month $j$, and soil depth $k$ ($\delta'\theta_{i,j,k}$) was computed as*

$$\delta'\theta_{i,j,k} = \theta_{i,j,k} - \theta_{hillslope,j,k} \quad , \tag{6}$$

*where $\theta_{hillslope,j,k}$ is the average soil moisture for all slope sites in month $j$ and soil depth $k$, as described above."*

$\delta$ was used in the section "3.2.1. Seasonal variability in soil moisture" to represent the "seasonal deviation" in soil moisture. Because $\delta\theta_{i,j,k}$ indicates the deviation in soil moisture $\theta_{i,j,k}$ from the annual average for that site and depth. $\delta'$ was introduced in section "3.2.2 Spatial variability in soil moisture at the hillslope scale" to represent the "spatial deviation" in soil moisture. In other words, $\delta'\theta_{i,j,k}$ in equation (6) indicates the deviation in the hillslope soil moisture $\theta_{i,j,k}$ from the hillslope average for that month and depth.

Line 224: why not other topographic properties such as slope?

Thanks for this suggestion. We have tried this in the data analysis process, but soil moisture and slope are not correlated well in our study area. The table below shows the Spearman rank correlations:

| Month | Rank correlation | p-value |
|---|---|---|
| 1 | -0.194 | 0.127 |
| 2 | -0.190 | 0.136 |
| 3 | -0.174 | 0.171 |
| 4 | -0.169 | 0.1840 |
| 5 | -0.120 | 0.349 |
| 6 | -0.050 | 0.698 |
| 7 | -0.125 | 0.329 |
| 8 | -0.021 | 0.872 |
| 9 | -0.183 | 0.150 |
| 10 | -0.157 | 0.220 |
| 11 | -0.197 | 0.121 |
| 12 | -0.180 | 0.158 |

Line 232: why not try cos(aspect) as it looks like a good indicator for soil moisture (https://doi.org/10.1016/j.jhydrol.2017.05.054)

**Thanks for this suggestion. We calculated the correlation between soil moisture and cos(aspect), but they are also not correlated well in our study area. The table below gives the Spearman rank correlations:**

| Month | Rank correlation | p-value |
|---|---|---|
| 1 | 0.029 | 0.823 |
| 2 | 0.024 | 0.855 |
| 3 | -0.022 | 0.867 |
| 4 | -0.131 | 0.306 |
| 5 | -0.012 | 0.923 |
| 6 | 0.039 | 0.758 |
| 7 | 0.018 | 0.888 |
| 8 | 0.126 | 0.326 |
| 9 | 0.002 | 0.986 |
| 10 | 0.084 | 0.511 |
| 11 | -0.178 | 0.162 |
| 12 | 0.029 | 0.818 |

**Instead of a correlation test, we linked the difference in monthly soil moisture and the difference in incoming solar radiation between the NW-facing and SE-facing slopes to determine the effect of aspect on soil moisture (Figs. 7 and 9).**

Figure 6c: title from y axis is missing. Can you please add measurement error for each graph?

**We added a title for the y-axis as "*P, PET, and P-PET [mm/month]*". Error bars were also added on each panel. We also fixed a few minor errors for the monthly P and PET calculation, resulting in slightly different values. This correction does not affect our conclusions.**

Line340-342: this is not that obvious visually. I would suggest authors to improve figure 7 by also making sure they are readable in white and black.

**If only white and black colors are used to represent dry and wet conditions, we would miss the representation of the seasonal changes in spatial variation (smaller spatial variation from May to July, larger spatial variation from October to March, which are now depicted with lighter and darker colors, respectively). What we can do is to add an aspect layer as a base map to help support the findings.**

Line 350-356: I don't think the convex-upward model is necessarily applicable to Chinese Loess Plateau. Even mean soil water content higher than 20% was measured, this pattern was not observed in a previous study (https://doi.org/10.1016/j.geoderma.2011.02.008). Authors may want to discuss a bit more here.

**We now include this reference and rewrote the discussion as follows: "*The most widely reported model for describing the relation between the spatial heterogeneity in soil moisture and mean soil moisture is a convex-upward parabola, with spatial variability peaking at intermediate values of soil moisture content (at approximately 20%) (Brocca et al., 2010; Famiglietti et al., 2008; Jarecke et al., 2021; Peterson et al., 2019; Tague et al., 2010; Western et al., 2003). This convex parabola has been observed in loess catchments as well (Gao et al., 2011; Gao et al., 2015; Shi et al., 2014), where spatial variability peaked at 15%-20% soil moisture content. In a similar loess system, Hu et al. (2011) found that the spatial variability increased slightly with increasing soil moisture, even in wetter conditions (20%-25%), indicating that a natural logarithmic curve might better describe the relationship between the spatial variability and average soil moisture. In the Gutun catchment, the average soil moisture was mainly between 5%-15%, which means that we may have observed only the short rising segment of a convex parabola below the variability peak, or the middle section of a logarithmic curve.*"**

Line 373-374: again, this is not that obvious though this may be true. Please consider to improve the figure 7. Probably rather than showing point value, it would be good idea to mapping the whole area?

**Although certain methods, such as Kriging interpolation, can map the soil moisture distribution for the entire catchment, they may generate artifacts and strange values in areas lacking monitoring sites. This would be misleading. Instead we prefer to present the actual measured values, which we believe are more reliable and informative. We added an aspect layer as a base map into Fig. 7 to better illustrate the findings.**

Line 405-406: the statistical analysis here can be misleading as less samples from gully than slope.

**The correlation analysis was specifically based on monitoring data from the hillslope sites, excluding the gully areas, for this reason. Therefore, our result indicates that there is no significant correlation between TWI and soil moisture patterns at the hillslope scale. We now clarify this point and highlight the "hillslope scale" in the revised manuscript: "*…We found no statistically significant correlation (α = 0.05) between TWI and soil moisture on the hillslopes for any soil depth, or averaged over the top 100 cm of the soils in each month. Note that we focus on the relationship between TWI or aspect with soil moisture patterns at the hillslope scale, excluding the gully. Soil moisture at the catchment scale is markedly higher in the gully (Figs. 2-3), consistent with the high TWI values there.*"**

Line 411-413: I would be cautious to draw this conclusion because gully was much wetter in this study.

**We now emphasize the "hillslope scale" in the conclusion: "*Therefore, as a typical proxy of topography, TWI is probably not a suitable index for explaining the soil moisture pattern on the hillslopes in such systems (Dymond et al., 2021)*"**

Line 426: the role of aspect in driving water variation was also documented in previous studies in the same areas. These papers need to be included in discussion.

**We now include several related studies conducted in the loess plateau in L383: "*…and thus more consistent soil moisture between the two slopes at the Gutun catchment. Hu et al. (2017) and Gao et al. (2016) showed a similar pronounced impact of aspect on soil moisture patterns in other catchments in loess landscapes.*"**

References:

*Hu, W., Shao, M., Han, F. and Reichardt, K., 2011. Spatio-temporal variability behavior of land surface soil water content in shrub-and grass-land. Geoderma, 162: 260-272.*

*Jia, X., Shao, M., Wei, X. and Wang, Y., 2013. Hillslope scale temporal stability of soil water storage in diverse soil layers. Journal of Hydrology, 498: 254-264.*

*Wang, Y., Shao, M., Zhu, Y. and Liu, Z., 2011. Impacts of land use and plant characteristics on dried soil layers in different climatic regions on the Loess Plateau of China. Agricultural and Forest Meteorology, 151: 437–448.*

*Wang, Y., Shao, M., Liu, Z. and Horton, R., 2013. Regional-scale variation and distribution patterns of soil saturated hydraulic conductivities in surface and subsurface layers in the loessial soils of China. Journal of Hydrology, 487: 13–23.*

**2) Author's response to the comments by Kendra Kaiser**

This study synthesizes information collected from a dense soil moisture monitoring network in a watershed with deep loess soils. They have highlighted the limitations of using the topographic wetness index for understanding the spatial variability of soil moisture in places that do not have a shallow confining layer, and that the spatial variability in these hillslopes is largely driven by evapotranspiration. The introduction is well written, while the discussion could use some editing to highlight the main findings and the more clearly share the associated story. Below in my specific comments I give suggestions on how to edit and trim the discussion to help focus on the most interesting findings.

**Thank you very much for these helpful comments. We have addressed all comments in detail below (in bold font).**

L23 – confusing since only part of the year is captured in these distinctions

**Based on the seasonal patterns in soil moisture (Figs. 6a and 6b), May to July and October to March were under dry and wet conditions, respectively. April was defined as the wet-to-dry transition month, and August and September were defined as the dry-to-wet transition months (L314-315) as for these months the deviation from the annual average is around 0 (Fig. 6b). These three months bridge the wet and dry conditions and it is challenging to precisely assign them to a specific wet or dry condition.**

L114 – Gully land consolidation --- I'm not sure if this is necessary here, but I am curious why this is being done and if it is occurring in other watersheds in the area.

**The Gully land consolidation is a project undertaken by the Chinese government in the Chinese Loess Plateau. With an investment of approximately 4.6 billion dollars, it aimed to address the problem of farmland loss caused by local reforestation activities. Starting in 2000, the Chinese government undertook a large-scale reforestation effort to address the severe erosion problem in the loess plateau. In this process, certain agricultural lands on the hillslopes were abandoned and dedicated to grass/tree planting or natural vegetation recovery. To prevent the adverse impact of farmland abandonment on local farmers' income, the gully land consolidation project was implemented alongside the reforestation: filling and levelling the gullies with soils from adjacent hillslopes, converting them into new farmland suitable for crop cultivation. As a consequence, the topography of the catchments has been altered compared to natural undisturbed ones, especially in the gully areas, making them into a newly typical and widespread catchment type in the loess plateau. Although our study was not primarily focused on the hydrological changes resulting from this project, we think it is necessary to acknowledge this point to distinguish such a disturbed catchment from natural ones.**

L187 – why the 20% trimmed standard deviation?

**We used this to exclude the effect of outliers on the calculated standard deviation but to avoid similar confusion for readers, we now use the non-trimmed standard deviation to calculate the temporal and spatial variabilities. Re-calculation using non-trimmed standard deviation gives similar results to our robust estimates of variability (20% trimmed standard deviations used now), thus it does not alter our conclusions. We updated the results in Fig. 4 and Fig. 8.**

L238 & 305 – suggest to edit sentences as to not start with "Fig. 2/6"

**We edited the sentence in L238: "_The seasonal changes in the average soil moisture for each depth for the hillslope ($\theta_{hillslope,j,k}$) and gully ($\theta_{gully,j,k}$) are shown in Fig. 2._"**

**We also edited the sentence in L305: "_The seasonal patterns in average soil moisture over the top 100 cm for the NW-facing slope, SE-facing slope, and gully sites ($\theta_{SE,j,0-100}$, $\theta_{NW,j,0-100}$,_**

and $\theta_{gully,j,0-100}$), and the deviations from their annual averages ($\delta\theta_{SE,j,0-100}$, $\delta\theta_{NW,j,0-100}$, and $\delta\theta_{gully,j,0-100}$), are illustrated in Fig. 5."

L290 – figure 4 caption is a little unclear, it seems as though the figure labels should be before the description, e.g. a) depth of max trimmed stdv, b) depth of collapse. How are 90% of sites "collapsing" between 20-100cm and 82% of sites collapsing between 160 and 260?

**We relocated the figure labels to appear before the corresponding descriptions for Fig. 4, also for the other figures.**

**Fig. 4 displays the tallied histogram representing the frequency distribution in the depth of maximum trimmed standard deviation (TSD) and the depth at which TSD collapses for 72 monitoring sites. The x-axis indicates the count of the monitoring sites. In panel (a), we observed that the maximum TSD occurred at a depth of 0-20 cm for 15 monitoring sites, at a depth of 20-40 cm for 20 sites, at 40-60 cm for 8 sites, at 60-80 cm for 17 sites, and at 80-100 cm for 5 sites. Therefore, for a total of 65 sites (=15+20+8+17+5), the maximum TSD occurred within the top 100 cm soil depth. As a percentage of the total number of sites (excluding those were measurements are truncated before 5m depth), this is 65/72 ≈ 90%. Therefore, we concluded that the depth of the maximum TSD was between 0 and 100 cm for 90% of the sites. By the same way, we concluded that the depth at which the TSD collapses was between 160 and 260 cm for 82% of the sites.**

**We changed the x-axis title from "_Frequency_" to "_Count of sites_" to enhance the clarity. The percentages are a bit different after the result have been updated (re-calculation by non-trimmed SD), but this will not affect our conclusions.**

L282-286 is a longer description of the content in L 316-321 which is easier to read and is associated with a mechanism, consider removing the former.

**We removed the content in L282-286. Additionally, we exchanged the order of Fig. 5 and Fig. 6, then included a simple sentence describing the tallied histograms to complement the main finding in L316-321: "…, with a wet-to-dry transition in April and a dry-to-wet transition between August and September (Figs. 5a-b). _Together with the tallied histograms of the months in which the annual maximum and annual minimum soil moisture occurred for each soil layer (Fig. 6),_ these results suggest that soil moisture in the top 100 cm of the soil was at a minimum in the late spring and early summer, …"**

L298 – What is the color scheme in Fig 5 for?

**We removed the color scheme and replaced it by a single color for Fig. 5a and 5b, respectively.**

Fig 6 could be easier to read as a vertically stacked figure so that months are aligned

**We agree and re-stacked the panels in a vertical way.**

L341 – figure color descriptions should be in a legend or in caption, not in the text

**We removed the figure color descriptions from L341 and added them in the caption of Fig. 7:** *"Deviations in soil moisture at 0-100 cm ($\delta'\theta_{i,j,0-100}$) from the monthly average soil moisture for the hillslope sites. The lighter colors indicate small deviations from the average soil moisture (values close to 0), while darker colors indicate larger deviations. Blue and red indicate soil moisture above and below the average, respectively."*

L 349 – this is slightly misleading as written given that the max soil moisture is below 20% in the hillslopes. You get there in the paragraph, but the set up suggests that there is something different happening here than in the literature.

**We revised this paragraph considering the comments from you and reviewer #1:** *"The most widely reported model for describing the relation between the spatial heterogeneity in soil moisture and mean soil moisture is a convex-upward parabola, with spatial variability peaking at intermediate values of soil moisture content (at approximately 20%) (Brocca et al., 2010; Famiglietti et al., 2008; Jarecke et al., 2021; Peterson et al., 2019; Tague et al., 2010; Western et al., 2003). This convex parabola has been observed in loess catchments as well (Gao et al., 2011; Gao et al., 2015; Shi et al., 2014), where spatial variability peaked at 15%-20% soil moisture content. In a similar loess system, Hu et al. (2011) found that the spatial variability increased slightly with increasing soil moisture, even in wetter conditions (20%-25%), indicating that a natural logarithmic curve might better describe the relationship between the spatial variability and average soil moisture. In the Gutun catchment, the average soil moisture was mainly between 5%-15%, which means that we may have observed only the short rising segment of a convex parabola below the variability peak, or the middle section of a logarithmic curve."*

Figure 7 –an aspect layer as a basemap to help support the findings re: aspect and moisture patterns would be helpful; label the legend and edit the caption to start with "Deviation in VWC from ...

**Thanks for this suggestion. We added an aspect layer as a basemap in Fig. 7.**
**The legend label is now "*Deviation in soil moisture*".**
**We also edited the caption:** *"Deviations in soil moisture at 0-100 cm ($\delta'\theta_{i,j,0-100}$) from the monthly average soil moisture for the hillslope sites."*

L374 – is this statement supported given the high amount of overlap between boxplots in Fig9? Is it statistically significant? There can be a pattern without being statistically significant, but it should be

clarified.

**We tested the significance of the differences with one-way ANOVA, the result is as follows:**

| Month | p-value | p-range | note |
|---|---|---|---|
| 1 | 0.000715 | p<0.001 | *** |
| 2 | 0.000773 | p<0.001 | *** |
| 3 | 0.000912 | p<0.001 | *** |
| 4 | 0.000009 | p<0.001 | *** |
| 5 | 0.0213 | p<0.05 | * |
| 6 | 0.0191 | p<0.05 | * |
| 7 | 0.0297 | p<0.05 | * |
| 8 | 0.626 | ns | ns |
| 9 | 0.000054 | p<0.001 | *** |
| 10 | 0.000092 | p<0.001 | *** |
| 11 | 0.703 | ns | ns |
| 12 | 0.00180 | p<0.01 | ** |

**For most of the months (except August and November), the differences in soil moisture between the two slopes were statistically significant (under dry conditions, in May to July, the difference is less distinct). We have noted the result for significance test in the Fig. 9(a) and its caption. We now further note it in L374: "*Fig. 7 also reveals a spatial pattern in soil moisture, with the NW-facing slope being much wetter than the SE-facing slope, which was also seen in Fig. 5a, and is quantified in more detail (including the significance test) in Fig. 9.*"**

L410-412 – this is the thing that I find most interesting and broadly relevant, but clarify this is referring to spatial variability in the hillslopes (not the hillslope vs gully)

**We clarified this as suggested: "*We found no statistically significant correlation (α = 0.05) between TWI and soil moisture on the hillslopes for any soil depth, or averaged over the top 100 cm of the soils in each month. Note that we focus on the relationship between TWI or aspect with soil moisture patterns at the hillslope scale, excluding the gully. Soil moisture at the catchment scale is markedly higher in the gully (Figs. 2-3), consistent with the high TWI values there.*"**

Section 4.3 – consider editing these first two paragraphs to be more integrated, e.g. L422-425 reads like results while L435-437 gets into the context that makes them interesting.

**We edited these two paragraphs to be more integrated.**

L450 – confusing as written – most of the precipitation fell between driest and wettest months?

**We removed the entire paragraph in which the L450 is located as suggested by the next comment.**

L449 – this paragraph isn't particularly compelling as written, it walks through calculations and results presented in the table. Move the table to results and retain the last sentence in the discussion.

**We removed this paragraph, and moved the last sentence into the first paragraph of Section 4.3: "*Spatial patterns of annual soil moisture storage change (ΔS) at depths of 0-100 cm, 100-200 cm, and 200-300 cm are illustrated in Fig. 10. The average ΔS that we measured in the 0-300 cm soils for the entire catchment was 110 mm. This is broadly similar to the water balance estimate (Table 1), suggesting that the top 300 cm of soil accounts for most or all of the seasonal water storage in the Gutun catchment.*"**

**We also fixed a few minor errors in the annual water balance estimation resulting in slight discrepancies in the revised manuscript. This correction does not affect our conclusions.**

**3) Author's response to the comments by Yanhui Wang**

This study explored the seasonal dynamics and spatial patterns of soil moisture by intensive measurements of soil water content from 5 m profile in a loess catchment in 2016-2021. The results have practical implications for catchment-scale hydrologic modeling and the design of soil moisture monitoring networks. The paper is well written, but there are a few minor issues that need improvement before acceptance.

**Thank you very much for the positive evaluation. We have addressed all comments in detail below (in bold font).**

Line 18: Isn't it 5.5 years from April 2016 to October 2021?

**Thank you for spotting this typo. We corrected 6.5 years to 5.5 years: "*In this study, volumetric soil moisture was monitored monthly for 5.5 years at 20 cm intervals between the surface and 500 cm depth at 89 sites across a small (0.43 km²) catchment on the Chinese Loess Plateau.*"**

Line 95-96: Please explain what local and nonlocal controls specifically denote.

**We have provided a comprehensive overview of the "local" and "non-local" controls in the first paragraph of the Introduction section, please see Lines 53-60: "*Grayson et al. (1997) and Western et al. (2003) demonstrated that topography has a greater influence on spatial patterns of soil moisture under wet conditions, due to the redistribution of soil water by lateral flow, resulting in wetter soils along hillslope drainage lines in convergent topography. Under dry conditions, by contrast, soil properties and vegetation become more important factors because soil moisture is*

*mainly affected by point-scale vertical water fluxes. Any topographic influence during dry conditions is more likely to be due to aspect rather than topographic convergence (Grayson and Western, 2001). Grayson and Western (2001) summarized this phenomenon as local and nonlocal controls on soil moisture under dry and wet conditions, respectively."*

Line 260: The title from x axis is missing in Fig. 2., and the x axis scale capital letter meaning should also be stated. Check the other figures in the paper in the same way.

**We added a note concerning the capital letters in the x-axis of Fig. 2: "*The capital letters on the x-axis indicate the months from January (J) to December (D).*" We believe the x-axis title can be omitted with the help of the new added notes.**

Line 262-264: The explanation of the data results should appear in the results and discussion rather than in the figure title. Other figure titles in the paper also have this problem, please modify it.

**We believe that providing an interpretation within the figure captions can improve the overall readability of the manuscript. Readers can directly grasp the main message conveyed by the figures without having to flip back and forth to the main text. Furthermore, many readers scan the figures of a paper without reading the entire text; putting the main points in the figure captions allows these readers to immediately grasp most of the main points of the paper. For these reasons, some academic journals require figures to have informative captions to enhance the overall quality of the manuscript. We therefore prefer to retain the explanatory content in figure captions, as part of a deliberate strategy that allows many readers to immediately understand what the figures mean, and to grasp many of the main points of the paper, even if they do not read the text.**

Line 402-404: "Some previous studies (e.g., Western et al. (2003) have reported that soil moisture patterns are predominantly shaped by topographic convergence, and that these effects are stronger during the wet season". The grammar of this sentence is wrong, please correct it.

**We corrected the grammar: "*Some previous studies (e.g., Western et al. (2003) have reported that soil moisture patterns are predominantly shaped by topographic convergence, and that this effect is stronger during the wet season.*"**

---

## Author Response (AR2)

We thank the reviewers for their comments on our manuscript. Please find below the reviewers' original comments (in plain text) and our responses (**in bold**).

Reviewer #1:

I think authors have successfully addressed my concerns expect they have misunderstood my comment "Line340-342: this is not that obvious visually. I would suggest authors to improve figure 7 by also making sure they are readable in white and black". I was not suggesting them to change figure 7 in white and black, but to make sure it is still readable if it is printed in white and black. It is a minor issue but worth paying attention to.

**Thank you for this helpful comment. We have changed the figure to use a divergent color scale. This color scale employs white in the center (values from -1 to 1) to represent minimal soil moisture deviations, light red on the left (values from -4 to -1) to indicate soil moisture below the average, and dark blue on the right (values from 1 to 4) to indicate soil moisture above the average. The red and blue are in light and dark, respectively, which can make the color scale readable when it is printed in black and white. In addition, we have given the symbols a black outline so that the points with a light color are more distinct from the gray background.**

Reviewer #3:

The authors presented a long-term dataset of soil moisture in a small catchment on the Chinese Loess Plataea. The manuscript is easy to read. Although no reviewers have posed serious questions per se during the open discussion phase, the novelties of this study are not still very clear. Here are few comments that, I hope, can be of any use for the authors.

**Thank you for these comments.**

1. The effect of North- and South-facing slopes (on soil moisture or other processes) is well studied in the past, particularly in eco-geomorphology. There is nothing really surprising about the results. The authors can refer to some of the refs I listed below. In my opinion, those studies performed much better analysis than the current study.

**Of course, we are also not surprised that south-facing slopes have a lower soil moisture content. However, we were surprised (and others may also be surprised) that the effects of aspect are largest during the winter, when ET rates are lowest. Regarding the first study mentioned by the reviewer, Gutiérrez-Jurado et al.'s results are potentially confounded by variations in soil characteristics. One major advantage of our loess study site is that there is almost no variation in soil properties from one location to the next; thus we are better able to isolate the effects of topography on soil moisture. This is particularly important for low moisture contents when texture has a relatively large effect on soil moisture content. We now also cite Gutiérrez-Jurado et al. (2007) at line 401, and explicitly make this point. Regarding the second study, Srivastava et al. (2021)'s**

**results are based purely on a modeling exercise. We do not understand how this can be considered a "much better analysis" than real-world data.**

**Note that in response to this comment, we have also changed the brightness of the aerial photograph in Figure 1b.**

2. The authors raised the question regarding local vs. nonlocal factors. First, the authors need to specify what local and nonlocal factors are. Second, it seems the authors did not really address this question in the end.

**In fact, we did specify what local and nonlocal factors are, at lines 55-63 of the previous version, but we have clarified the text to make this more obvious. We have also mostly removed the local/nonlocal distinction, in favor of a more explicit contrast between aspect-controlled ET variations (a local control) and downslope flow controlled by hillslope convergence (a nonlocal control) but now also refer back to these terms explicitly on L419 and 421 in the discussion.**

3. For the topographic effect, the authors need to consider what are the spatial and temporal scales that under consideration. Similar to the spatial scale effect, the topographic impact can be very different at different temporal scales (e.g., storm event vs. mean annual), as the controlling processes can vary significantly.

**We agree. However, we hope it is obvious that we have presented a seasonal analysis based on monthly data, so we have no information about soil moisture responses at the storm-event timescale. Similarly, it should be clear that we focus on the hillslope and catchment scale. However, we now state this explicitly on L111 above the research questions. Note that we addressed this difference in scale explicitly by looking at the hillslopes and gully sites separately, and also in the discussion on L424-431. We prefer not to comment on spatial or temporal scales for which we have no data.**

4. For deep soil moisture analysis, another important issue is the traveling time between surface and depth, particularly under dry conditions (due to low unsaturated hydraulic conductivity). Sometimes, it may take several months or years for infiltrated water to reach deep soil layers. This should be considered in the analysis.

**We agree that deep infiltration can take long time spans, particularly in arid regions. However, there is so little deep infiltration below 2-3 meters at our site (see Figure 3a) that we have no practical way to observe these long timescales in our data.**

5. Given the long-term data presented here, I highly recommend the authors to go deeper than the current analysis, which basically provides no new information other than another case study and does not really provide very important implications as claimed by the authors.

**We strongly disagree with the characterization of our work as just "another case study". We have analyzed an unusually dense network (89 locations in a 43 hectare catchment) of unusually deep measurements (up to 5 m depth), in a loess catchment whose very uniform substrate allows us to analyze topographic controls on seasonal soil moisture patterns. This work yields surprising results, such as that soil moisture is highest during the months of least precipitation, and vice versa. Another interesting result was that the topographic wetness index was a very poor predictor of**

moisture patterns, despite its widespread use in models. We now highlight these novel aspects more clearly in the abstract, introduction and conclusion (L16-20, L91-94, L477, and L481).